# Structured Neural Networks for Density Estimation and Causal Inference

**Asic Chen**[1*]   **Ruian Shi**[1*]   **Xiang Gao**[1]   **Ricardo Baptista**[2†]   **Rahul G. Krishnan**[1†]

[1]University of Toronto, Vector Institute    [2]California Institute of Technology

{asicchen, ruiashi, xgao, rahulgk}@cs.toronto.edu
rsb@caltech.edu

## Abstract

Injecting structure into neural networks enables learning functions that satisfy invariances with respect to subsets of inputs. For instance, when learning generative models using neural networks, it is advantageous to encode the conditional independence structure of observed variables, often in the form of Bayesian networks. We propose the Structured Neural Network (StrNN), which injects structure through masking pathways in a neural network. The masks are designed via a novel relationship we explore between neural network architectures and binary matrix factorization, to ensure that the desired independencies are respected. We devise and study practical algorithms for this otherwise NP-hard design problem based on novel objectives that control the model architecture. We demonstrate the utility of StrNN in three applications: (1) binary and Gaussian density estimation with StrNN, (2) real-valued density estimation with Structured Autoregressive Flows (StrAFs), autoregressive normalizing flows that leverage StrNN as a conditioner, and (3) interventional and counterfactual analysis with StrAFs. Our work opens up new avenues for learning neural networks that enable data-efficient generative modeling and the use of normalizing flows for causal effect estimation.

## 1  Introduction

The incorporation of structure into machine learning models has been shown to provide benefits for model generalization, learning efficiency, and interpretability. The improvements are particularly salient when learning from small amounts of data. This idea has found use in reinforcement learning [Ok et al., 2018], computational healthcare [Hussain et al., 2021, Cui et al., 2020], survival analysis [Gharari et al., 2023], time series analysis [Curi et al., 2020], and causal inference [Balazadeh et al., 2022].

This work focuses on the problem of density estimation from high-dimensional data which has been approached through a variety of lenses. For example, normalizing flows [Tabak and Turner, 2013, Rezende and Mohamed, 2015] model data by transforming a base distribution through a series of invertible transformations. Masked autoencoders (MADE) [Germain et al., 2015] model the joint distribution via an autoregressive factorization of

$$p(X) = p_{\theta_1}(x_1)p_{\theta_2}(x_2|x_1)p_{\theta_3}(x_3|x_2)p_{\theta_4}(x_4|x_2,x_3)$$

Figure 1: StrNN injects structure by masking the weights of a neural network. *Top*: StrNN connections (green) compared to a fully connected network (gray). *Bottom*: Binary factorization of an adjacency matrix yields weight masks. Masked weights shown in gray.

---

[*]Equal Contribution [†]Equal Senior Authorship

37th Conference on Neural Information Processing Systems (NeurIPS 2023).

the random variables. The factorization is enforced by integrating structure in the neural network of the autoencoder. The MADE architecture zeros out weights in a neural network to ensure each output dimension has an autoregressive dependence on the input dimensions. When data is scarce, this may lead to over-fitting and harm generalization. When knowledge of a Bayesian network [Pearl, 2011] and the associated conditional independencies exist, it is desirable to inject this knowledge directly into the network to improve density estimation. In this work, we extend the concept of weight masking, as in MADEs, to go beyond autoregressive dependencies of the output on the input.

This work proposes the **Structured Neural Network (StrNN)**, a network architecture that enforces functional independence relationships between inputs and outputs via weight masking. In other words, the output can remain unaffected by changes to (subsets of) the input. We focus on instantiating this idea to model conditional independence between inputs when neural networks are deployed to model the density of random variables, as illustrated in Figure 1. Any set of conditional independence statements (e.g. in a Bayesian network) may be represented via a binary adjacency matrix. StrNN performs binary matrix factorization to generate a set of weight masks that follow the adjacency matrix. There are two key challenges we overcome. First, the general problem of binary matrix factorization in this context is under-specified, as there exist many valid masks whose matrix product realizes a given adjacency matrix. To this end, we propose the idea of neural network path maximization as a strategy to guide the generation of optimal masks. Secondly, binary matrix factorization is NP-hard in general. We study practical solutions that generate valid mask matrices efficiently.

StrNN can then be applied to NN-based density estimation in various contexts. Where conditional independence properties are known a priori, we show that StrNN can be used to estimate parameters of data distributions while keeping specified variables conditionally independent. We further integrate StrNNs into various discrete and continuous flow architectures, including the autoregressive normalizing flow [Papamakarios et al., 2017, Huang et al., 2018] to form the **Structured Autoregressive Flow (StrAF)**. The StrAF model uses the StrNN as a normalizing flow conditioner network, thus enforcing a given adjacency structure within each flow layer. The StrAF preserves variable orderings between chained layers, allowing the adjacency structure to be respected throughout the entire flow.

Finally, we study a natural application of StrAF in causal effect estimation, a domain that often requires flexible methods for density estimation that enforce conditional independence relationships within causal graphs [Pearl, 2009]. We show how StrAFs can be used to perform interventional and counterfactual queries better than existing flow-based causal models that do not incorporate graphical structures. Across the board, we highlight how incorporating conditional independence structure improves generalization error when learning from a small number of samples.

In summary, the main contributions of this work are as follows:

1. We introduce StrNN, a weight-masked neural network that can efficiently learn functions with specific variable dependence structures. In particular, it can inject prior domain knowledge in the form of Bayesian networks to probability distributions. We formalize the weight masking as an optimization problem, where we can pick the objective based on desired neural architectures. We propose an efficient binary matrix factorization algorithm to mask arbitrary neural networks.

2. We integrate StrNN into autoregressive and continuous normalizing flows for best-in-class performance in density estimation and sample generation tasks.

3. We apply StrAF for causal effect estimation and showcase its ability to outperform existing causal flow models in accurately addressing interventional and counterfactual queries.

## 2   Background

**Masked Autoencoders for Density Estimation (MADE):** Masked neural networks were introduced for density estimation on binary-valued data [Germain et al., 2015]. Given $\mathbf{x} = (x_1, ..., x_d)$, MADE factorizes $p(\mathbf{x})$ as the product of the outputs of a neural network. Writing the $j$-th output as the conditional probability $\hat{x}_j := p(x_j = 1|\mathbf{x}_{<j})$, the joint distribution can be rewritten exactly as the binary cross-entropy loss. As long as the neural network outputs are autoregressive in relation to its inputs, we can minimize the cross-entropy loss for density estimation. To enforce the autoregressive property for a neural network $y = f(x)$ with a single hidden layer and $d$ inputs and outputs, MADE element-wise multiplies the weight matrices W and V with binary masks $M^W$ and $M^V$:

$$h(x) = g((W \odot M^W)x + b), \quad y = f((V \odot M^V)h(x) + c). \tag{1}$$

The autoregressive property is satisfied as long as the product of the masks, $M^V M^W \in \mathbb{R}^{d \times d}$, is lower triangular. The MADE masking algorithm (2) can be extended to neural networks with an arbitrary number of hidden layers and hidden sizes. For Gaussian data, the MADE model can be extended as $\mathbb{R}^d \rightarrow \mathbb{R}^{2d}$, $(x_1, ... x_d) \rightarrow (\hat{\mu}_1, ..., \hat{\mu}_d, \log(\hat{\sigma}_1), ..., \log(\hat{\sigma}_d))$. The last mask must be duplicated to ensure $\mu_j$ and $\sigma_j$ only depend on $\mathbf{x}_{<j}$. MADE can also be used as the conditioner in an autoregressive flow to model general data, as seen in Papamakarios et al. [2017].

**Normalizing Flows:** Normalizing flows [Rezende and Mohamed, 2015] model complex data distributions and have been applied in many scenarios [Papamakarios et al., 2021]. Given data $\mathbf{x} \in \mathbb{R}^d$, a normalizing flow $\mathbf{T} \colon \mathbb{R}^d \rightarrow \mathbb{R}^d$ takes $\mathbf{x}$ to latent variables $\mathbf{z} \in \mathbb{R}^d$ that are distributed according to a simple base distribution $p_{\mathbf{z}}$, such as the standard normal. The transformation $\mathbf{T}$ must be a diffeomorphism (i.e., differentiable and invertible) so that we can compute the density of $\mathbf{x}$ via the change-of-variables formula: $p_{\mathbf{x}}(\mathbf{x}) = p_{\mathbf{z}}(\mathbf{T}(\mathbf{x})) |\det J_{\mathbf{T}}(\mathbf{x})|$. We can compose multiple diffeomorphic transformations $\mathbf{T}_k$ to form the flow $\mathbf{T} = \mathbf{T}_1 \circ \cdots \circ \mathbf{T}_K$ since diffeomorphisms are closed under composition. The flows are trained by maximizing the log-likelihood of the observed data under the density $p_{\mathbf{x}}(\mathbf{x})$. The log-likelihood can be evaluated efficiently when it is tractable to compute the Jacobian determinant of $\mathbf{T}$; for example when $\mathbf{T}_k$ is a lower triangular function [Marzouk et al., 2016]. Given the map, we can easily generate i.i.d. samples from the learned distribution by sampling from the base distribution $\mathbf{z}^i \sim p_{\mathbf{z}}$ and evaluating the flow $\mathbf{T}^{-1}(\mathbf{z}^i)$.

**Density Estimation with Autoregressive Flows:** When the Jacobian matrix of each flow layer is lower triangular, its determinant is simply the product of its diagonal entries [Huang et al., 2018]. This gives rise to the **autoregressive flow** formulation: given an ordering $\pi$ of the $d$ variables in the data vector $\mathbf{x}$, the $j$th component of the flow $\mathbf{T}$ has the form: $x_j = \tau_j(z_j; c_j(\mathbf{x}_{<\pi(j)}))$ where each $\tau_j$ is an invertible *transformer* and each $c_j$ is a *conditioner* that only depends on the variables that come before $x_j$ in the ordering $\pi$. As a result, the map components define an autoregressive model that factors the density over a random variable $x$ as: $p(\mathbf{x}) = \prod_{j=1}^{d} p(x_j|\mathbf{x}_{<j})$ where $\mathbf{x}_{<j} = (x_1, \dots, x_{j-1})$. *autoregressive*. Under mild conditions, any arbitrary distribution $p_{\mathbf{x}}$ can be transformed into a base distribution with a lower triangular Jacobian [Rezende and Mohamed, 2015]. That is, autoregressive flows are arbitrarily expressive given the target distribution. One common choice of invertible functions for the transformer are monotonic neural networks [Wehenkel and Louppe, 2019].

**Density Estimation with Continuous Normalizing Flows:** Continuous normalizing flows (CNFs) [Chen et al., 2018, Grathwohl et al., 2018] represent the transformation $\mathbf{T}$ as the flow map solving the differential equation $\frac{\partial \mathbf{z}(t)}{\partial t} = f(\mathbf{z}(t), t; \theta)$. Given the initial condition $\mathbf{x} = \mathbf{z}(t_1) \sim p_{\mathbf{x}}$, we can integrate $f$ backwards in time from $t_1$ to $t_0$ to obtain $\mathbf{z} = \mathbf{z}(t_0) \sim p_{\mathbf{z}}$, or vice versa. In order to learn the CNF, Chen et al. [2018] computes the change in log-density under the transformation using the *instantaneous change of variables* formula, which is defined by the differential equation $\frac{\partial \log p(\mathbf{z}(t))}{\partial t} = -\text{Tr}(\frac{\partial f}{\partial \mathbf{z}(t)})$. This expression is used to compute the log-likelihood of a target sample as $\log p(\mathbf{z}(t_1)) = \log p(\mathbf{z}(t_0)) - \int_{t_0}^{t_1} \text{Tr}(\frac{\partial f}{\partial \mathbf{z}(t)}) dt$. We refer the reader to Chen et al. [2018] for the process of back-propagating through the objective using the adjoint sensitivity method as well as a discussion on the existence and uniqueness of a solution for the ODE flow map.

**Causal Inference with Autoregressive Flows:** Modelling causal relationships is crucial for enabling effective decision-making in various fields [Pearl, 2009]. A structural equation model (SEM) parameterizes the process that generates observed data, allowing us to reason about interventions and counterfactuals. Given random variables $\mathbf{x} = (x_1, ..., x_d) \in \mathbb{R}^d$ with joint distribution $\mathbb{P}_{\mathbf{x}}$, the associated SEM consists of $d$ structural equations of the form $x_j = f_j(\mathbf{pa}_j, u_j)$, where $u$ represents mutually independent latent variables and $\mathbf{pa}_j$ denotes the direct causal parents of variable $x_j$. Each SEM also corresponds to a directed acyclic graph (DAG), with a causal ordering $\pi$ defined by the variables' dependencies.

Khemakhem et al. [2021] showcased the intrinsic connection between SEMs and autoregressive flows. The authors demonstrated that affine autoregressive flows with a fixed ordering of variables can be used to parameterize SEMs under the framework of *causal autoregressive flows* (CAREFL). When the causal ordering of variables is given, CAREFL outperforms other baselines in interventional tasks and generates accurate counterfactual samples. However, one significant limitation of CAREFL is that there is no guarantee that the autoregressive structure corresponds to the true dependencies in the causal graph beyond pairwise examples. In Section 3.3, we leverage StrAF to incorporate additional independence structure, enhancing its faithfulness to the causal DAG.

## 3 Methodology

### 3.1 Structured Neural Networks

As neural networks are universal function approximators [Hornik et al., 1989], injecting structure into multilayer perceptrons (MLPs) naturally allows us to approximate complex functions with certain invariances. For a neural network that approximates an arbitrary function $f : \mathbb{R}^m \to \mathbb{R}^n$, one type of invariance we could consider is when a specific output $\hat{x}_j$ is independent from a given input $x_i$, i.e.: $\hat{x}_j \perp x_i$ means $\frac{\partial \hat{x}_j}{\partial x_i} = 0$. As a motivating example, we focus on the probabilistic density estimation problem framed as learning maps from one probability distribution to another. In this setting, it is imperative that we are able to learn structured functions between two high-dimensional spaces. Therefore in this paper, we seek to efficiently encode such invariances via weight masking.

For data $\mathbf{x} = (x_1, ..., x_d)$, we use the lower-triangular adjacency matrix $A \in \{0, 1\}^{d \times d}$ to represent the underlying variable dependence structure. In other words, $A_{ij} = 0$ for $j < i$ if and only if $x_i \perp x_j | x_{\{1,...,i\} \setminus j}$ and $A_{ij} = 1$ otherwise. This matrix encodes the same information as a Bayesian network DAG of the variables. In the fully autoregressive case, matrix A is a dense lower triangular matrix with all ones under the diagonal, which is the only case addressed in Germain et al. [2015]. Their proposed MADE algorithm (Appendix A.1) only encodes the structure of dense adjacency matrices, and cannot incorporate additional conditional independencies. Further, the non-deterministic version of the algorithm can introduce unwanted independence statements by chance, as discussed in Proposition 1.

We improve upon the idea of masked autoregressive neural networks to directly encode the independence structure represented by an adjacency matrix $A$ that is lower triangular but also has added sparsity. We observe that a masked neural network satisfies the structural constraints prescribed in $A$ if the product of the masks for each hidden layer has the same locations of zero and non-zero entries as $A$. Therefore, given the conditional independence structure of the underlying data generating process, we can encode structure into an autoregressive neural network by *factoring the adjacency matrix into binary mask matrices for each hidden layer*.

More specifically, given an adjacency matrix $A \in \{0, 1\}^{d \times d}$ and a neural network with $L$ hidden layers, each with $h_1, h_2, ..., h_L$ hidden units ($\geq d$), we seek mask matrices $M^1 \in \{0, 1\}^{h_1 \times d}, M^2 \in \{0, 1\}^{h_2 \times h_1}, ..., M^L \in \{0, 1\}^{d \times h_L}$ such that $A' \sim A$, where $A' := M^L \cdots M^2 \cdot M^1$. We use $A' \sim A$ to denote that matrices $A'$ and $A$ share the same sparsity pattern, i.e.: exact same locations of zeros and non-zeros. Note that here $A$ is a binary matrix and $A'$ is an integer matrix. We then mask the neural network's hidden layers using $M^1, M^2, ..., M^L$ as per (1) to obtain a **Structured Neural Network (StrNN)**, which respects the prescribed independence constraints. The value of each entry $A'_{ij}$ thus corresponds to the number connections flowing from input $x_j$ to output $\hat{x}_i$ in the StrNN.

Finding the optimal solution to this problem is NP-hard since binary matrix factorization can be reduced to the biclique covering problem [Miettinen and Neumann, 2020, Ravanbakhsh et al., 2016, Orlin, 1977]. Furthermore, most existing works focus on deconstructing a given matrix $A$ into low-rank factors while minimizing (but not eliminating) reconstruction error [Dan et al., 2015, Fomin et al., 2020]. In our application, any non-zero reconstruction error breaks the independence structure we want to enforce in our masked neural network. This puts existing algorithms for low-rank binary matrix factorization outside the scope of our paper. We instead consider the problem of finding factors that reproduce the adjacency matrix exactly, which is always possible when hidden layer dimensions are greater than the input and output dimension.

**Optimization Objectives.** Identifiability remains an issue even when we eliminate reconstruction error. Given an adjacency matrix $A$, there can be multiple solutions for factoring $A$ into per-layer masks that satisfy the constraints, especially if the dimensions of the hidden layers are much larger than $d$. Since the masks dictate which connections remain in the neural network, the chosen mask factorization algorithm directly impacts the neural network architecture. Hence, it is natural to explicitly specify a relevant objective to the neural network's approximation error during the matrix factorization step, such as the test log-likelihood.

Given that the approximation error is inaccessible when selecting the architecture, we were inspired by the Lottery Ticket Hypothesis [Frankle and Carbin, 2018] and other pruning strategies [Srivastava et al., 2014, Gal and Ghahramani, 2016] that identify a subset of valuable model connections. Our hypothesis is that given the same data and prior knowledge on independence structure, the masked

neural network with *more connections* is more expressive, and will thus be able to learn the data better and/or more quickly. To find such models, we consider two objectives: Equation (2) that maximizes the number of connections in the neural network while respecting the conditional independence statements dictated by the adjacency matrix, and Equation (3) that maximizes connections while penalizing any pair of variables from having too many connections at the cost of the others. That is,

$$\max_{A' \sim A} \sum_{i=1}^{d} \sum_{j=1}^{d} A'_{ij}, \qquad (2) \qquad\qquad \max_{A' \sim A} \sum_{i=1}^{d} \sum_{j=1}^{i} A'_{ij} - \mathrm{var}(A'), \quad (3)$$

where $\mathrm{var}(A')$ is the variance across all entries in $A'$. While we focus on these two objectives, future work will find optimal architectures by identifying other objectives to improve approximation error.

**Factorization Algorithms.** The maximization of the aforementioned mask factorization is an intractable optimization problem. We therefore develop approximate algorithms to solve them. Our strategy for optimization involves recursively factorizing the mask matrix layer by layer. Given $A \in \{0,1\}^{d \times d}$, we run Algorithm 1 once to find $A_1 \in \{0,1\}^{d \times h_1}$ and $M^1 \in \{0,1\}^{h_1 \times d}$ for a layer of width $h_1$ such that $A_1 \cdot M^1 \sim A$, where $\sim$ denotes that the matrices share the same sparsity. For the next layer with $h_2$ hidden units, we use $A_1$ in place of $A$ to find $A_2 \in \{0,1\}^{d \times h_2}$ and $M^2 \in \{0,1\}^{h_2 \times h_1}$ such that $A_2 \cdot M^2 \sim A_1$. We repeat until we have found all the masks.

For each objective, we can obtain per-layer exact solutions using integer programming. While the Gurobi optimizer [Gurobi Optimization, LLC, 2023] can be used for small $d$, this approach was found to be too computationally expensive for $d$ greater than 20, which is a severe limitation for real-world datasets and models. We hereby propose a greedy algorithm (shown in **Algorithm 1**) that *approximates* the solution to the maximum connections objective in Equation (2). For each layer, the algorithm first replicates the structure given in the adjacency matrix $A$ by copying its rows into the first mask. It then maximizes the number of neural network connections by filling in the second mask with as many ones as possible while respecting the sparsity in $A$. See Appendix A.2 for a visual explanation of the algorithm. For a network with $d$-dimensional inputs and outputs and one hidden layer with $h$ units, this algorithm runs in $\mathcal{O}(dh)$ time. Scaling up to $L$ layers, where each hidden layer commonly contains $\mathcal{O}(d)$ units, the overall runtime is $\mathcal{O}(d^2 L)$, which is much more efficient than the integer programming solutions. From our experiments, the greedy algorithm executes nearly instantaneously for dimensions in the thousands.

In Appendix A, we include detailed results from investigating the link between the neural network's generalization performance and the choice of mask factorization algorithm. We observe that while the exact solution to objective (2) achieves a higher objective value than the greedy approach, it has no clear advantage in density estimation performance. Moreover, we found that models trained with the two objectives, (2) and (3), provide similar performance. However, some datasets might be more sensitive to the exact objective. For example, problems with anisotropic non-Gaussian structure may require neural architectures with more expressivity in some variables that may be favored with certain objectives. While we adopted objective (2) and the efficient greedy algorithm in the remainder of our experiments, designing and comparing different factorization objectives is an important direction for future work.

---

**Algorithm 1:** Greedy factorization

---

**Data:** $A \in \{0,1\}^{d_1 \times d_2}$, hidden size h
**Result:** $M^V \in \{0,1\}^{d_1 \times h}$,
$\qquad\qquad M^W \in \{0,1\}^{h \times d_2}$, satisfying
$\qquad\qquad M^V M^W \sim A$

1 $nz \leftarrow$ non-zero rows in $A$
2 Fill $M^W$ with $nz$; repeat until full
3 Fill $M^V$ with ones
4 for $i$-th row in $M^V$ do:
5 $\quad C \leftarrow$ indices of 0's in $i$-th row of $A$
6 $\quad T \leftarrow$ cols. of $M^W$ whose index $\in C$
7 $\quad R \leftarrow$ indices of non-zero rows of $T$
8 $\quad$ for $r$ in $R$: set $M^V_{i,r}$ to zero
9 return $(M^V, M^W)$

---

### 3.2 Structured Neural Networks for Normalizing Flows

For the general real-valued probabilistic density estimation task, we highlight the use of StrNN in two popular normalizing flow frameworks, autoregressive flows and continuous-time flows. In the autoregressive flows setting, recall that the $j$th component of each flow layer is parameterized as $x_j = \tau_j(z_j; c_j(\mathbf{x}_{<\pi(j)}))$, where the conditioner $c_j$ dictates which inputs the latent variable can depend on. We use the StrNN as the conditioner, combined with any valid invertible transformer $\tau$, to form the **Structured Autoregressive Flow (StrAF)**. The StrNN conditioner ensures each transformed latent

variable is only conditionally dependent on the subset of variables defined by a prescribed adjacency matrix. A single step from StrAF can be represented as: $\mathbf{z} = \text{StrAF}(\mathbf{x}) = \tau(\mathbf{x}, \text{StrNN}(\mathbf{x}, A))$ where $\mathbf{x} \in \mathbb{R}^d$ and $A \in \{0,1\}^{d \times d}$ is an adjacency matrix. Normalizing flow steps are typically composed to improve the expressiveness of the overall flow. To ensure that the prescribed adjacency structure is respected throughout all flow steps, we simply avoid the common practice of variable order permutation between flow steps. We visualize this in Figure 2. This choice does not hurt performance as long as a sufficiently expressive transformer is used. In our experiments, we apply the unconstrained monotonic neural network (UMNN) described in Wehenkel and Louppe [2019] as a transformer.

We also integrate the StrNN into continuous normalizing flows (CNFs), where the neural network that parameterizes $f$ in the differential equation $\frac{\partial \mathbf{z}(t)}{\partial t} = f(\mathbf{z}(t), t)$ is replaced by a StrNN to obtain $\frac{\partial \mathbf{z}(t)}{\partial t} = \text{StrNN}(\mathbf{z}(t), t, A)$. This plug-in replacement allows us to inject structure into the function describing the continuous dynamics of the CNF without modifying other aspects of the CNF. We refer to this architecture as the **Structured CNF (StrCNF)**. The StrCNF uses the trace estimator described in FFJORD [Grathwohl et al., 2018] to evaluate the objective function for learning the flow.

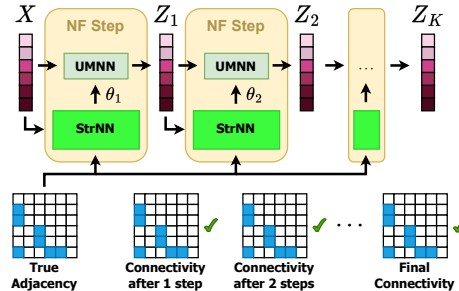

**Structured Autoregressive Flow (StrAF)**

Figure 2: The StrAF injects a prescribed adjacency into each flow step using a StrNN conditioner. The StrAF does not permute latent variables, allowing the adjacency matrix to be respected throughout the entire flow.

While injecting structure, both StrAF and StrCNF inherit the efficiency of StrNN due to our choice of weight masking. Specifically, the output of the StrNN can be computed with a single forward pass through the network. In comparison, input masking approaches such as Wehenkel and Louppe [2021] must perform $d$ forward passes to compute the output for a single datum. This not only prevents efficient application of input masking to high dimensional data, but also is a barrier to integrating the method with certain architectures. For example, the CNF already requires many neural network evaluations to numerically solve the ODE defining the flow map, so making $d$ passes per evaluation is particularly inefficient.

### 3.3 Structured Causal Autoregressive Flow

Beyond density estimation, StrAF allows us to build autoregressive flows that faithfully represent variable dependencies defined by a causal DAG. In contrast, CAREFL [Khemakhem et al., 2021] only maintains the autoregressive order, which is insufficient for data with more than two variables where the true structure must be characterized by a full adjacency matrix. Building on CAREFL, we also assume that the flow $\mathbf{T}$ takes on the following affine functional forms for observed data $\mathbf{x}$:

$$x_j = e^{s_j(\mathbf{x}_{<\pi(j)})} z_j + t_j(\mathbf{x}_{<\pi(j)}), \quad j = 1, ..., d \tag{4}$$

where the functions $s_j$ and $t_j$ are both conditioners that control the dependencies on the variables preceding $x_j$. It is crucial for the autoregressive or graphical structures to be maintained across all sub-transformations $\mathbf{T}_1, ..., \mathbf{T}_K$ for the flow $\mathbf{T} = \mathbf{T}_1 \circ \cdots \circ \mathbf{T}_K$.

Assuming the true causal topology has been given either from domain experts or oracle discovery algorithms, StrAF can directly impose the given topological constraints by adding an additional masking step based on the adjacency matrix, as shown in section 3.1. This ensures that the dependencies of the flows match the known causal structure, which leads to more accurate inference predictions.

## 4 Related Works

Given a Bayesian network adjacency matrix, Wehenkel and Louppe [2021] introduced graphical conditioners to the autoregressive flows architecture through input masking. They demonstrated that unifying normalizing flows with Bayesian networks showed promise in injecting domain knowledge while promoting interpretability, as even single-step graphical flows yielded competitive results in density estimation. Our work follows the same idea of introducing prior domain knowledge into

autoregressive flows, but we instead use a masking scheme similar to methods in Germain et al. [2015]. We again note that input masking scales poorly with data dimension $d$, as $d$ forward passes are required to obtain the output for a single datum.

Silvestri et al. [2021] proposed embedded-model flows, which alternates between traditional normalizing flows layers and gated structured layers that a) encode parent nodes based on the graphical model, and b) include a trainable parameter that determines how strongly the current node depends on its parent nodes, which alleviates error when the assumed graphical model is not entirely correct. In comparison, our work encodes conditional independence more directly in the masking step, improving accuracy when the assumption in the probabilistic graphical structure is strong.

Mouton and Kroon [2022] applied a similar idea to residual flows by masking the residual blocks' weight matrices prior to the spectral normalization step according to the assumed Bayesian network. Similarly, Weilbach et al. [2020] introduced graphical structure to continuous normalizing flows by masking the weight matrices in the neural network that is used to parameterize the time derivative of the flow map. However, their method is difficult to apply to neural networks with more than a single hidden layer. In comparison, our factorization algorithm more naturally permits the use of multi-layered neural networks when representing CNF dynamics. The Zuko software package [Rozet et al., 2023] implements various types of normalizing flows, including MAFs that also rely on weight masking, but they only provide one possible algorithm to enforce autoregressive structure given a specific variable order. In comparison, our approach permits explicit optimization of different objectives during the adjacency matrix factorization step and we investigate the efficacy of factorization schemes and resulting neural architectures in our work. For completeness, we include pseudocode of their algorithm and comparisons to our own algorithms in Appendix A .

Flows have garnered increasing interest in the context of causal inference, with applications spanning various problem domains. Ilse et al. [2021] parameterized causal model with normalizing flows in the general continuous setting to learn from combined observational and interventional data. Melnychuk et al. [2022] used flows as a parametric method for estimating the density of potential outcomes from observational data. Flows have also been employed in causal discovery [Brouillard et al., 2020, Khemakhem et al., 2021] as well as in various causal applications [Ding et al., 2023, Wang et al., 2021]. In particular, Balgi et al. [2022a] also considered embedding the true causal DAG in flows for interventional and counterfactual inference, but they do so via the framework of Graphical Normalizing Flows [Wehenkel and Louppe, 2021]. Balgi et al. [2022b] used CAREFL on a real-world social sciences dataset leveraging a theorized Bayesian network.

## 5 Experiments

To demonstrate the efficacy of encoding structure into the learning process, we show that using StrNN its flow integrations to enforce a prescribed adjacency structure improves performance on density estimation and sample generation tasks. We experiment on both synthetic data generated from known structure equations and MNIST image data. Details on the data generation process for all synthetic experiments can be found in Appendix C. We also apply StrAF in the context of causal inference and demonstrate that the additional graphical structure introduced by StrAF leads to more accurate interventional and counterfactual predictions. The code to reproduce these experiments is available at https://github.com/rgklab/StructuredNNs.

### 5.1 Density Estimation on Binary Data

For binary density estimation, we compare StrNN against the fully autoregressive MADE baseline.

**Synthetic Tabular Data** We generate binary tabular data through structural equations from known Bayesian networks. The results are shown in Figure 3 (left). We find that StrNN performs better than MADE, especially in the low data regime, as demonstrated on the left hand side of each chart.

**MNIST Image Data** To study the effect of structure in image modeling, we use the binarized MNIST dataset considered in Germain et al. [2015], Salakhutdinov and Murray [2008]. Germain et al. [2015] treated each 28-by-28-pixel image as a 784-dimensional data vector with full autoregressive dependence. Since we do not know the ground truth structure, we use StrNN to model a local autoregressive dependence on a square of a pixels determined by the hyper-parameter nbr_size. By changing the hyperparameter we can increase the context window used to model each pixel.

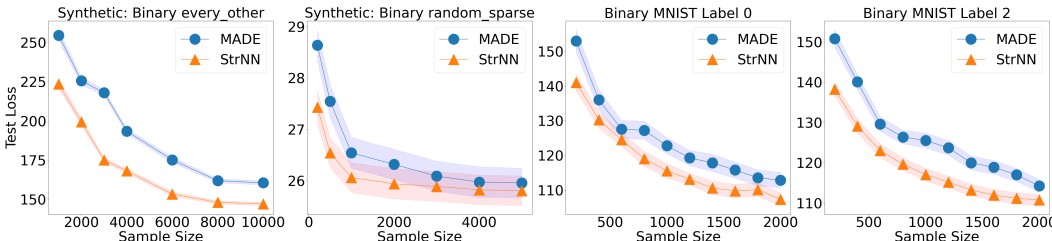

Figure 3: Negative log likelihood of a test set (lower is better) for density estimation experiments on binary synthetic data (left 2 images) and label-dependent binary MNIST data (right 2 images). Error ranges are reported as standard error across the test set. StrNN performs better than MADE as the sample size decreases.

StrNN is equivalent to MADE for this experiment when we set `nbr_size=28`. We first find the best `nbr_size` for each label via grid search. For labels 0 and 2, the optimal `nbr_size` is 10, and per-label density estimation results can be found in Figure 3 (right). StrNN outperforms MADE for both labels, with the advantage more significant when sample size is very small. Samples of handwritten digits generated from both StrNN and MADE can be found in Appendix B.

## 5.2 Density Estimation on Gaussian Data

We run experiments to compare the performances of StrNN and MADE on synthetic Gaussian data generated from known structure equation models where each $x_i$ is Gaussian. We plot the results in Figure 4. StrNN achieves lower test loss than MADE on average, although the error bars are not necessarily disjoint. When the sample size is low, however, StrNN significantly outperforms MADE, similar to the binary case. In conclusion, across all binary and Gaussian experiments, *encoding structure* makes StrNN significantly more accurate at density estimation than the fully autoregressive MADE baseline.

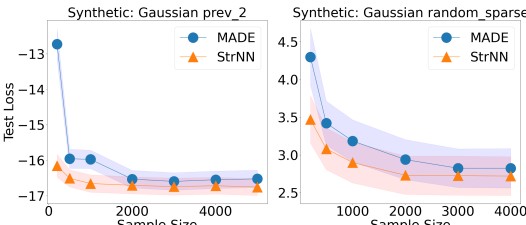

Figure 4: Results from density estimation experiments on Gaussian synthetic data generated from two different sparsity patterns. Test loss is reported in negative log likelihood with error ranges (standard error across test set). StrNN performs significantly better than MADE in the low data regime, and better on average.

## 5.3 Density Estimation with Structured Normalizing Flows

We evaluate StrAF on density estimation against several baselines. We draw 1000 samples from a 15 dimensional tri-modal and non-linear synthetic dataset for experimental evaluation. Data generation is further described in Appendix C.4.

**Experimental Setup** We select the fully autoregressive flow (ARF) and the Graphical Normalizing Flow (GNF) [Wehenkel and Louppe, 2021] as the most relevant discrete flow baselines for comparison. We use the official GNF code repository during evaluation, but note that it contains design decisions that harm sample quality (see Appendix E.3). We also evaluate against FFJORD [Grathwohl et al., 2018] and Weilbach et al. [2020] as baselines for StrCNF. While other structured flows exist and have been examined in Section 4, they do not represent variables using an autoregressive structure and hence are less comparable. Where applicable, models were provided the true adjacency matrix in their conditioners. All discrete flow models use a UMNN [Wehenkel and Louppe, 2019] transformer and we grid-search other hyperparameters as described in Appendix E.2. We evaluate density estimation performance using the negative log-likelihood (NLL). After fixing the hyperparameters, we perform 8 randomly initialized training runs, then report the mean and 95% CI of the test NLL for these runs.

**StrNN improves flow-based models for density estimation**

We report results in Table 1 and observe several trends. For both discrete and continuous flows, the ability to incorporate structure yields performance benefits compared to the ARF and FFJORD baselines. In particular, the StrNN offers advantages in comparison to baseline approaches that can encode structure. For example, while the method proposed by Weilbach et al. [2020] allows a DAG structure to

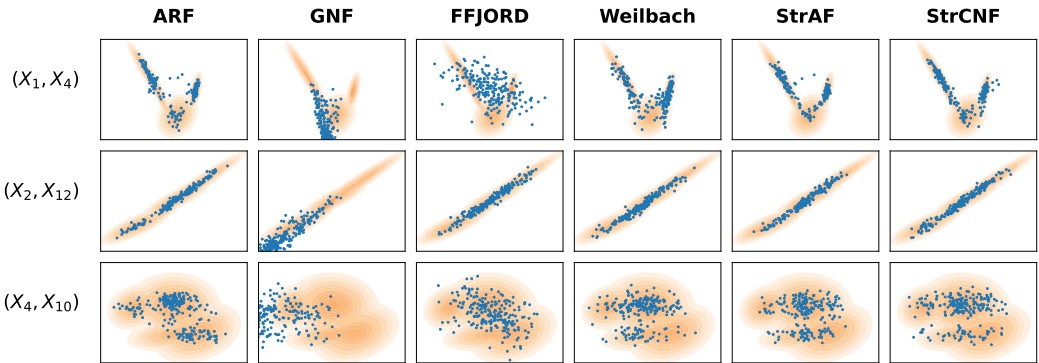

Figure 5: Model generated samples are shown in blue dots for randomly selected dimensions. The ground truth density is visualized by the orange contours. See Appendix E.3 for an explanation on why GNF performs poorly.

be injected into the baseline CNF, its inability to easily use a multi-layered neural network to represent dynamics hinders performance as compared to the StrCNF. We observe that StrAF and GNF perform comparably, and the StrCNF outperforms all other models. We visualize the quality of samples generated by these flow models in Figure 5, and find that both StrAF and StrCNF yields samples that closely match the ground truth distribution.

Table 1: Evaluation of flow-based models. Mean and 95% CI of test NLL over 8 runs reported.

|  | Test NLL ($\downarrow$) |
| --- | --- |
| ARF | -3.09 $\pm$ 0.43 |
| GNF | **-3.63 $\pm$ 0.35** |
| StrAF | **-3.55 $\pm$ 0.20** |
| FFJORD | -1.85 $\pm$ 0.64 |
| Weilbach | -2.59 $\pm$ 0.58 |
| StrCNF | **-4.01 $\pm$ 0.12** |

## 5.4 Causal Inference with Structured Autoregressive Flows

We conduct synthetic experiments where we generate data according to a linear additive SEM. Details on data generation can be found in Appendix D.2. In our experiments involving 5- and 10-variable SEMs, we compare StrAF against CAREFL, which only utilizes MADE as the conditioner and maintains autoregressive ordering of the variables. This comparison highlights the additional benefits of enforcing the generative model to be faithful to the causal graph, which is a feature unique to StrAF. Furthermore, unlike the previous work conducted by Khemakhem et al. [2021] that evaluates causal queries on individual variables alone, we propose a comprehensive evaluation metric called total mean squared error (MSE) for these two causal tasks as outlined below. We report the mean errors along with the standard deviations from multiple training runs with different datasets. The formulations of the metrics can be found in Appendix D.3. In addition, Appendix D.1 provides detailed algorithms and related discussions on generating interventional samples and computing counterfactuals with flows.

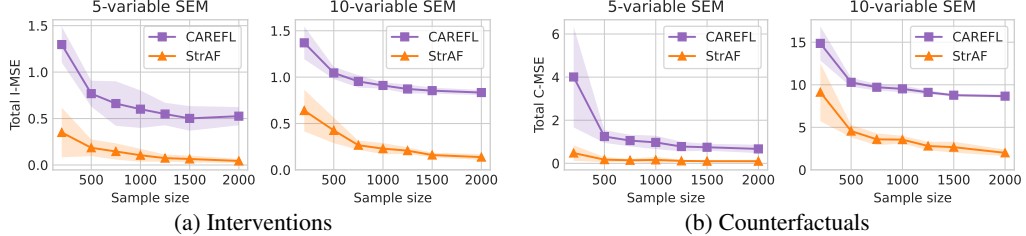

(a) Interventions         (b) Counterfactuals

Figure 6: Evaluations of causal predictions (**left**: interventions; **right**: counterfactuals) on 5- and 10-variable SEMs made by StrAF and CAREFL. Performance is measured by the corresponding total mean squared error with standard deviation across ten runs. (a) measures the error of the expected value of one variable under different interventions, while (b) computes the error by deriving counterfactual values under different observed samples and queries.

**Interventions.** We perform interventions on each variable $x_j$ within the SEM and draw samples from the intervened causal system. Interventions are performed by setting the intervened value $\alpha$ using one of eight integers perturbed from the mean of the intervened variable. We compute the expectations $\mathbb{E}[x_i|do(x_j = \alpha)]$ for variable $x_i$, excluding the intervened variable $x_j$ itself and its

preceding variables as they remain unaffected. To estimate these expectations, we use 1000 samples and calculate the squared error for the predictions. The resulting error is then averaged over the set of intervened values, intervened variables, and the corresponding variables with computed expectations under interventions. We present the plot of this aggregated metric, referred to as the total intervention mean squared error (total I-MSE), for the two SEMs in Figure 6a. Notably, StrAF consistently outperforms CAREFL across all training dataset sizes, highlighting the effectiveness of the additional graph structure in improving StrAF's inference of the interventional distribution.

**Counterfactuals.** In a similar setting as the intervention experiments, we evaluate StrAF and CAREFL on their ability to compute accurate counterfactuals conditioned on observed data. Counterfactual inference tackles what-if scenarios: determining the value of variable $x_i$ if variable $x_j$ had taken a different value $\alpha$. Unlike interventions, counterfactual queries involve deriving the latent variables $\mathbf{z}$ given the observed data $\mathbf{x} = \mathbf{x}_{\text{obs}}$, rather than sampling new $\mathbf{z}$. We generate 1000 observations using the synthetic SEM and derive counterfactual values $\mathbf{x}$ by posing queries with varying $\alpha$ values for each variable $x_j$. Similar to interventions, we compute the squared error and average over the 1000 observed samples and all possible combinations of counterfactual queries for each observed sample, and we refer to this metric as the total counterfactual mean squared error (total C-MSE). Figure 6b illustrates that, for both SEMs, StrAF outperforms CAREFL in making more accurate counterfactual predictions. Moreover, StrAF demonstrates consistent performance even in scenarios with limited available samples.

## 6   Conclusions and Limitations

We introduce the **Structured Neural Network (StrNN)**, which enables us to encode functional invariances in arbitrary neural networks via weight masking during learning. For density estimation tasks where the true dependencies are expressed via Bayesian networks (or adjacency matrices), we show that StrNN outperforms a fully autoregressive MADE model on synthetic and MNIST data. We integrate StrNN in autoregressive and continuous flow models to improve both density estimation and sample quality. Finally, we show that our structured autoregressive flow-based causal model outperforms existing baselines on causal inference. We address some limitations and directions for future work below.

**Access to true adjacency structure.** We assume access to the true conditional independence structure. While this information is available in many contexts, such as from domain experts, there is also a wide literature on learning the conditional independencies directly from data [Drton and Maathuis, 2017]. One prominent example is the NO-TEARS algorithm [Zheng et al., 2018]. Wehenkel and Louppe [2021] shows that integrating NO-TEARS with an autoregressive flow can improve density estimation when a ground truth adjacency is unknown. Any adjacency matrix learned from data can also be integrated in StrAF. In our work, we have found two existing causal structure discovery libraries that are relatively comprehensive and easy to use: Kalainathan et al. [2020] and Zheng et al. [2023]. Future work will use StrNN to directly learn structure from data, providing a full pipeline from structure discovery to density estimation and sample generation.

**StrNN optimization objectives.** The mask factorization algorithm used by StrNN can maximize different objectives while ensuring the matrices satisfy a sparsity constraint. In Section 3.1, we proposed two such objectives and in Appendix A we demonstrated that they can impact model generalization. StrNN provides a framework with which it is possible to explore other objectives to impose desirable properties on neural network architectures. Moreover, investigating the effect of sparse structure on faster and easier training is a valuable direction [Frankle and Carbin, 2018]. For example, one approach would be to leverage connections between dropout and the lottery ticket hypothesis to randomly introduce sparsity into a neural network using StrNN weight masking.

## Acknowledgments and Disclosure of Funding

We thank David Duvenaud, Tom Ginsberg, Vahid Balazadeh-Meresht, Phil Fradkin, and Michael Cooper for insightful discussions and draft reviewing. We thank anonymous reviewers for feedback that has greatly improved the work. This research was supported by an NSERC Discovery Award RGPIN-2022-04546, and a Canada CIFAR AI Chair. AC is supported by a DeepMind Fellowship and RS is supported by the Ontario Graduate Scholarship and the Ontario Institute for Cancer Research. Resources used in preparing this manuscript were provided in part, by the Province of Ontario, the Government of Canada through CIFAR, and companies sponsoring the Vector Institute.

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
