# Appendix

## Table of Contents

The code used in this project can be found at: `https://github.com/rgklab/StructuredNNs`.

## A  Mask Algorithms and Binary Matrix Factorization

### A.1  MADE Algorithm and Limitations

As mentioned in the main text, the central idea of masking neural network weights to inject variable dependence was inspired by the work of Germain et al. Germain et al. [2015]. In their paper, the authors proposed Algorithm 2 to ensure the outputs of an autoencoder are autoregressive with respect to its inputs.

As a concrete example, let us consider the case of a single hidden layer network with $d$ inputs and $h$ hidden units. Here, Algorithm 2 first defines a permutation $\mathbf{m}^0 \in \mathbb{R}^d$ of the set $\{1, \ldots, d\}$, and then independently samples each entry in the vector $\mathbf{m}^1 \in \mathbb{R}^h$ with replacement from the uniform distribution over the integers from 1 to $d - 1$. This assignment is used to define the two binary masks matrices $M^1$ of size $h \times d$ and $M^2$ of size $d \times h$. The matrix product of the resulting masks $M^2 M^1 \in \mathbb{R}^{d \times d}$ provides the network's connectivity. In particular, the $(i, j)$ entry of $M^2 M^1$ indicates the dependence of output $i$ on input $j$.

There are several key limitations of the MADE algorithm:

1. As mentioned in Section 3.1, the MADE algorithm can only mask neural networks such that they respect the autoregressive property. It is not capable of enforcing additional conditional independence statements as prescribed by an arbitrary Bayesian network. For a general probability distribution that does not satisfy any conditional independence properties, we expect each marginal conditional in the factorization of the density $p(x) = \prod_{k=1}^{d} p(x_k | x_{<k})$ to depend on all previous

---

**Algorithm 2:** MADE Masking Algorithm

---

**Input** : Dimension of inputs $d$, Number of hidden layers $L$, Number of hidden units $h$
**Output** : Masks $M^1, \ldots, M^{L+1}$

1  % Sample $\mathbf{m}^l$ vectors;
2  $\mathbf{m}^0 \leftarrow \text{shuffle}([1, \ldots, d])$;
3  **for** $l = 1$ *to* $L$ **do**
4     **for** $k = 1$ *to* $h^l$ **do**
5        $\mathbf{m}^l(k) \leftarrow \text{Uniform}([\min(\mathbf{m}^{l-1}), \ldots, d-1])$;
6     **end**
7  **end**
8  % Construct masks matrices;
9  **for** $l = 1$ *to* $L$ **do**
10    $M^l \rightarrow \mathbb{1}_{\mathbf{m}^l \geq \mathbf{m}^{l-1}}$;
11  **end**
12  $M^{L+1} \rightarrow \mathbb{1}_{\mathbf{m}^0 > \mathbf{m}^L}$;

---

    inputs. As a result, the matrix product $M^2 M^1$ should be fully lower-triangular, meaning that output $k$ depends on all inputs $1, \ldots, k-1$. If there is conditional independence structure, however, the MADE algorithm does not provide a mechanism to define mask matrices such that their matrix product is sparse and hence the corresponding MADE network enforces these constraints on the variable dependence.

2. The non-deterministic MADE masking algorithm presented in Germain et al. [2015], the resulting mask matrices are not always capable of representing any distribution. In particular, the random algorithm can yield some mask matrices where the lower-triangular part of their matrix product is arbitrarily sparse, i.e., there exists some $k < k'$ such that $(M^2 M^1)_{k,k'} = 0$. As a result, the MADE network with these masks enforces additional conditional independencies that are not necessarily present in the underlying data distribution. Proposition 1 formalizes this point.

**Proposition 1.** *There is a non-zero probability that Algorithm 2 will yield masks that enforce unwanted conditional independencies.*

*Proof.* For a single hidden layer ($L = 1$) neural network with $h$ units, there is a probability $1/d^h > 0$ of sampling $\mathbf{m}^1 = \mathbf{1}$, i.e., each entry is independently sampled to be 1. This vector yields a mask matrix $M^1$ that only has one non-zero column of ones at the index $k$ where $\mathbf{m}^0(k) = 1$. As a result, the matrix product $M^2 M^1$ also has only one non-zero column at index $k$, meaning that all outputs $k' > k$ depend only on $x_k$ and not on other input variables. Therefore, these mask matrices enforce the constraints $X_{k'} \perp\!\!\!\perp X_{<k} | X_k$ for all $k'$. Equivalently, a distribution that does not satisfy these constraint can not be represented using this MADE network. $\square$

In our work, we adopt a weight masking scheme by solving a binary matrix factorization problem that overcomes these limitations. Both the globally optimal and approximate solutions proposed in our paper are deterministic. Thus, we enforce all conditional independence properties exactly and ensure unwanted variable independence statement do not appear in our neural networks.

### A.2   Mask Factorization: Integer Programming and Greedy Algorithm

In Section 3.1, we showed that finding the weight masks for each neural network layer is equivalent to factoring the adjacency matrix that represents the input and output connectivity of these layers. We can find exact solutions to this problem by solving the optimization problem given in (2). This problem can be formalized as the following integer programming problem:

$$\text{Inputs: } A \in \{0,1\}^{d \times d} \tag{5}$$

$$\text{Outputs: } M^V \in \{0,1\}^{d \times h}, M^W \in \{0,1\}^{h \times d}$$

$$\max \sum_{i=1}^{d} \sum_{j=1}^{d} v_i w_j$$

$$\text{such that } v_i w_j > 0 \text{ if } A_{ij} = 1$$

$$v_i w_j = 0 \text{ if } A_{ij} = 0$$

$$\text{where } M^V = \begin{pmatrix} v_1 \\ v_2 \\ \dots \\ v_d \end{pmatrix} \text{ and } v_i \in \{0,1\}^{1 \times h}$$

$$\text{and } M^W = \begin{pmatrix} w_1 & w_2 & \dots & w_d \end{pmatrix} \text{ and } w_j \in \{0,1\}^{h \times 1}$$

To formulate a similar problem for the objective given in (3) instead, we simply replace the integer programming objective with

$$\max \left( \sum_{i=1}^{d} \sum_{j=1}^{d} v_i w_j - \text{Var}_{i,j}(v_i w_j) \right). \tag{6}$$

We used the Gurobi optimizer Gurobi Optimization, LLC [2023] to solve the above integer programming problems in our experiments, and found that exact solutions are found reasonably quickly for up to $d = 20$. For dimensions larger than 20, however, directly solving the integer programming problem becomes prohibitively expensive even on computing clusters with multiple GPUs, so it is intractable to seek exact solutions to these problems for most real-world datasets. Therefore, in this work we propose Algorithm 1, a greedy method that gives an approximate solution to the problem 5 very efficiently. Figure 7 provides a visual example of the steps performed by Algorithm 1.

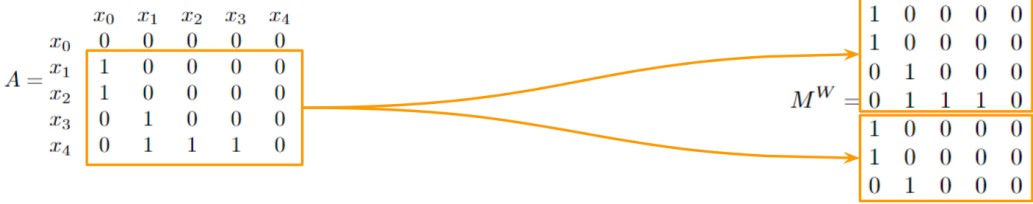

(a) Step 1: Given the adjacency matrix $A$, we first populate the first layer mask $M^W$ by copying over non-zero rows in $A$, and repeating until all rows of $M^W$ are full.

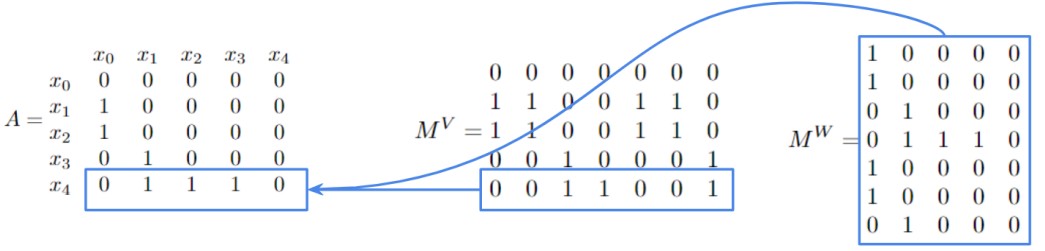

(b) Step 2: Next, we populate the second layer mask $M^V$. Let us take the last row of $M^V$ for an example: to respect all conditional independence statements given by $A$, we need the product of the last row of $M^V$ and the $M^W$ matrix to have the same zero and non-zero locations as the last row of $A$. Since there are zeros in the first and last column of $A$'s last row, we need the products of the last row of $M^V$ with the first and last columns of $M^W$ to be zero.

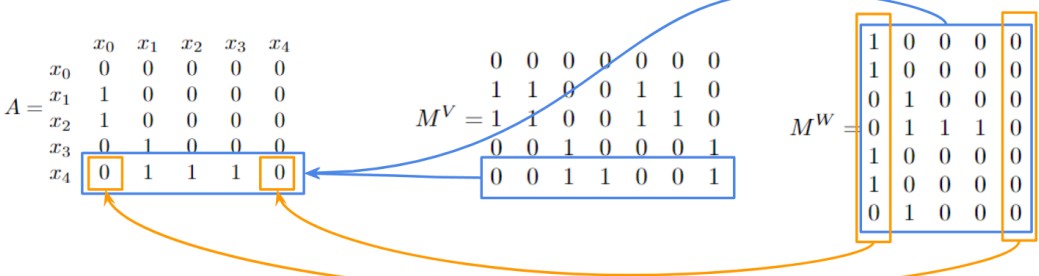

(c) Step 3: We find the unique ones in the first and last columns of $M^W$ and set the corresponding positions in the last row of $M^V$ to zero.

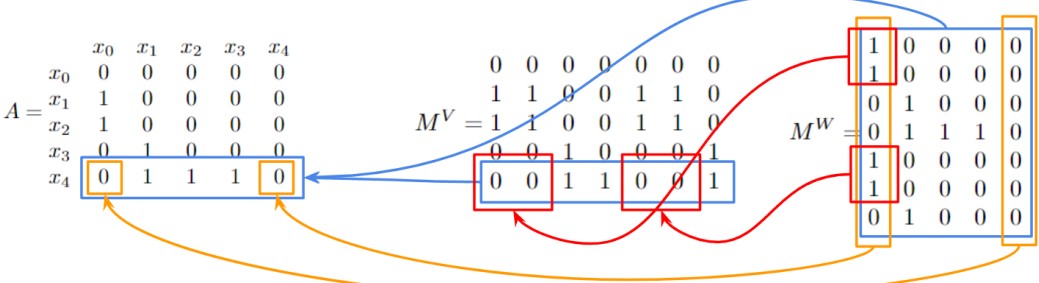

(d) Step 4: Everything else in the last row of $M^V$ is set to 1 to maximize the number of connections in the resulting neural network for the optimization objective in (2).

Figure 7: A visual example of Algorithm 1 being applied on the adjacency matrix A for a neural network with $d = 5$ inputs, $d = 5$ outputs, and one single hidden layer containing $h = 7$ hidden units.

To estimate how well the greedy algorithm approximates the solution to problem 5, we randomly sample lower triangular adjacency matrices, setting entries to 0 or 1 based on a given sparsity threshold between 0 and 1. In other words, for the threshold 0.1, the random adjacency matrix is very dense, and when the threshold is 0.9, it is very sparse. For fixed input and output dimensions, we sample 10 such random adjacency matrices for each sparsity threshold ranging from 0.1 to 0.9, and take the average of the $\sum_{i=1}^{d} \sum_{j=1}^{d} v_i w_j$ objective value obtained by each factorization algorithm. Results for $d = 10$-dimensional adjacency matrices are shown in Figure 8. We see that the exact integer programming solution achieves higher objective values compared to the greedy algorithm we propose in Algorithm 1, but it remains to further evaluate the resulting masks from both algorithms on their performance for a downstream density estimation task.

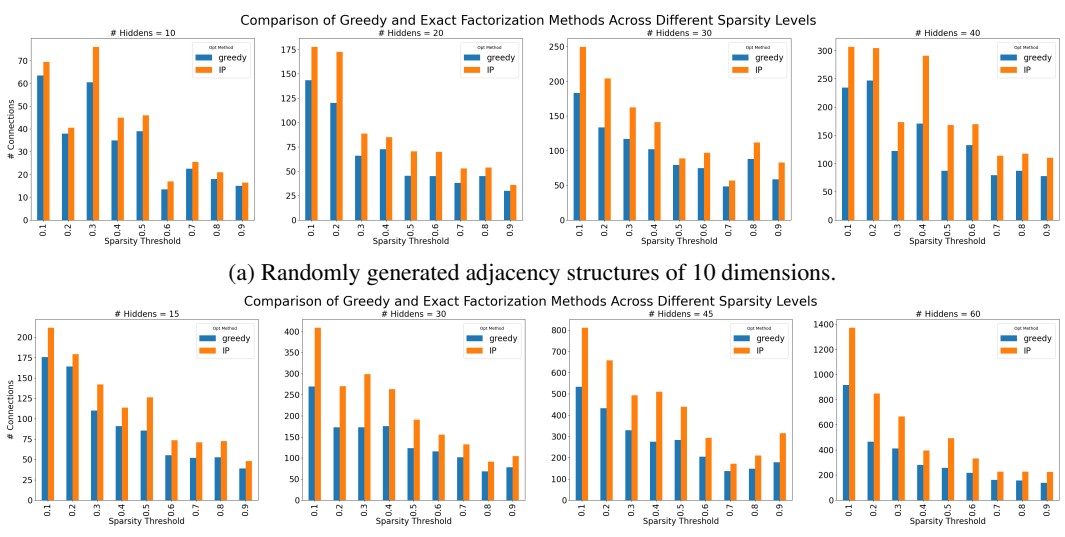

(a) Randomly generated adjacency structures of 10 dimensions.

(b) Randomly generated adjacency structures of 15 dimensions.

Figure 8: Comparing objective value (Equation 2) achieved by greedy and exact integer programming (IP) methods. IP gives better objective values when the adjacency matrix is very sparse. As the number of neurons involved goes up, the difference in these methods also increases.

## A.3   Mask Algorithms and Generalization

To that end, we evaluate the density estimation performance of masked neural networks on 20-dimensional synthetic binary data, using the following four mask factorization methods:

1. **MADE**: A fully autoregressive baseline using Algorithm 2 as proposed in Germain et al. [2015]

2. **Greedy**: The proposed method in Algorithm 1

3. **IP**: The exact integer programming solution to Problem 5

4. **IP-var**: The exact integer programming solution to Problem 5, but with the objective in (6)

5. **Zuko**: The approximate factorization algorithm proposed in [Rozet et al., 2023] Github repository, as per our understanding. (See Algorithm 3 for pseudocode and Appendix A.3.1 for a more detailed description of the Zuko method and its comparison to Algorithm 1.)

The experiment setup and grid of hyperparameters used for these experiments are the same as those in all binary and Gaussian experiments in this work, as detailed in Appendix E.1. Specifically, the adjacency structures used in these experiments are explained and visualized in C.1 and C.2. The results for the negative log-likelihood of a test dataset are reported in Table 2. We see that all three methods proposed and referenced in this work—Greedy, IP, IP-var, and Zuko—outperform the MADE baseline, but there is no clear winner based on overlapping error ranges. IP does not perform significantly better than Greedy based on the higher objective value achieved for (2). Meanwhile, there is no significant difference between the objective that maximizes the total connections (Equation 2 and the objective with the added variance penalty (Equation 3) when comparing the performance of

**Algorithm 3:** Zuko Masking Algorithm

---

**Input** : A: $\{0,1\}^{d\times d}$: adjacency matrix,
H = $(h_1, ..., h_{L-1}) \in \mathbb{N}^{L-1}$: list of $L-1$ hidden layer sizes
**Output** : Masks $M_1 \in \{0,1\}^{h_1\times d}, M_2 \in \{0,1\}^{h_2\times h_1}, \ldots, M_L \in \{0,1\}^{d\times h_{L-1}}$,
satisfying $M_L \times \cdots \times M_2 \times M_1 \sim A$

1 $A' \leftarrow$ unique rows of $A$;
2 $inv \leftarrow$ mapping of rows of $A'$ to the original row indices in $A$;
3 $P' \leftarrow A' \times A'.T$;
4 $n\_deps \leftarrow$ sums of rows of $A'$;
5 $P \leftarrow P' == n\_deps$;
6 **for** $i = 1$ *to* $L - 1$ **do**
7 $\quad$ **if** $i = 0$ **then**
8 $\quad\quad$ $M_i \leftarrow A'$;
9 $\quad$ **else**
10 $\quad\quad$ $M_i \leftarrow$ rows of $P$ denoted by $indices$;
11 $\quad$ **end**
12 $\quad$ **if** $i < L - 1$ **then**
13 $\quad\quad$ $reachable \leftarrow n\_deps \neq 0$;
14 $\quad\quad$ $indices \leftarrow reachable$ indices repeated to fill up to $h_i$;
15 $\quad\quad$ $M_i \leftarrow M_i[indices]$;
16 $\quad$ **else**
17 $\quad\quad$ $M_i \leftarrow M_i[inv]$
18 $\quad$ **end**
19 **end**

---

IP versus IP-var. Hence, for efficiency and overall performance, we choose to adopt the Greedy mask factorization algorithm for the rest of the experiments in this paper.

Table 2: Density estimation results on 20-dimensional synthetic datasets, reported using the negative log-likelihood on a held-out test set (lower is better). The error reported is the sample error across the test set. The four methods, Greedy, IP, IP-var, and Zuko perform better than the MADE baseline, but similarly to each other.

| Dataset | Random Sparse | | Previous 3 | |
| --- | --- | --- | --- | --- |
| | $n = 5000$ | $n = 2000$ | $n = 5000$ | $n = 2000$ |
| MADE | $7.790 \pm 0.140$ | $7.788 \pm 0.142$ | $8.767 \pm 0.132$ | $8.816 \pm 0.134$ |
| Greedy | $7.758 \pm 0.137$ | $7.778 \pm 0.142$ | $8.757 \pm 0.131$ | $\mathbf{8.768 \pm 0.130}$ |
| IP | $7.758 \pm 0.138$ | $7.769 \pm 0.140$ | $\mathbf{8.755 \pm 0.132}$ | $8.769 \pm 0.129$ |
| IP-var | $\mathbf{7.757 \pm 0.137}$ | $\mathbf{7.768 \pm 0.140}$ | $8.758 \pm 0.132$ | $8.770 \pm 0.131$ |
| Zuko | $7.758 \pm 0.137$ | $7.776 \pm 0.141$ | $8.757 \pm 0.130$ | $\mathbf{8.768 \pm 0.129}$ |

| Dataset | Every Other | |
| --- | --- | --- |
| | $n = 5000$ | $n = 2000$ |
| MADE | $8.373 \pm 0.120$ | $8.364 \pm 0.124$ |
| Greedy | $8.334 \pm 0.125$ | $8.315 \pm 0.123$ |
| IP | $8.333 \pm 0.129$ | $\mathbf{8.314 \pm 0.123}$ |
| IP-var | $\mathbf{8.331 \pm 0.126}$ | $\mathbf{8.314 \pm 0.124}$ |
| Zuko | $8.334 \pm 0.126$ | $8.315 \pm 0.123$ |

### A.3.1 Approximate Algorithm Comparisons

In this section, we give a more direct example of comparison on autoregressive flow performance between the greedy and Zuko algorithms. While the two factorization schemes of the adjacency matrix are similar in that they approximately maximize the number of remaining paths in the neural network, they can yield different results in specific settings. In the experiment below, mask matrices found by our Algorithm 1 and by Zuko (Algorithm 3) result in different performance. Zuko operates on the

unique rows of an adjacency matrix, which causes issues when the matrix contains many repeating rows. This type of structure may be naturally encountered in datasets with star shaped graphs. Consider a $d$-by-$d$ adjacency matrix where the first $(d-1)$ rows contain dependence on the first variable (i.e: $[1, 0, \ldots, 0]$) and only the last row depends on the second variable (i.e.: $[0, 1, 0, \ldots, 0]$). Given a budget of $h$ hidden units, Zuko would assign $\frac{h}{2}$ units to represent the variable corresponding to the last row. This ineffectively represents the dependence of all other outputs on the first variable, and is avoided by our greedy algorithm. In a non-linear dataset with $d$ variables and the above adjacency matrix, we compare StrAF with our greedy algorithm and the Zuko algorithm in the table below. We report runs from 5 random seeds, using a $95\%$ CI and $d$ hidden units. We see our method performs better, especially as $d$ increases. This further supports our claim that the choice of mask factorization can impact performance beyond enforcing a given independence structure.

Table 3: Comparison of StrAF performance using our greedy algorithm and the Zuko algorithm on a star-shaped Bayesian network. Runs from 5 random seeds are reported, using a $95\%$ CI and $d$ hidden units. Evaluation is based on negative log-likelihood on a held-out test set (lower is better)

| Dataset | Star-Shaped | |
| | $d = 50$ | $d = 1024$ |
| --- | --- | --- |
| Greedy | **0.4311± 0.5405** | **-18.734 ± 1.999** |
| Zuko | $1.3276 \pm 0.2806$ | $-14.550 \pm 0.936$ |

# B    Additional Experiment Results

To validate the sample generation quality of StrNN when trained on the MNIST dataset, we display select samples generated by both StrNN and the MADE baseline in Figure 9. We show that even in the low data regime (models fitted on 1000 training samples), both models generate samples with reasonable quality. Hence, we observe that for the density estimation task, injecting prior structure using a StrNN improves likelihood values for each sample under the model without sacrificing generative quality.

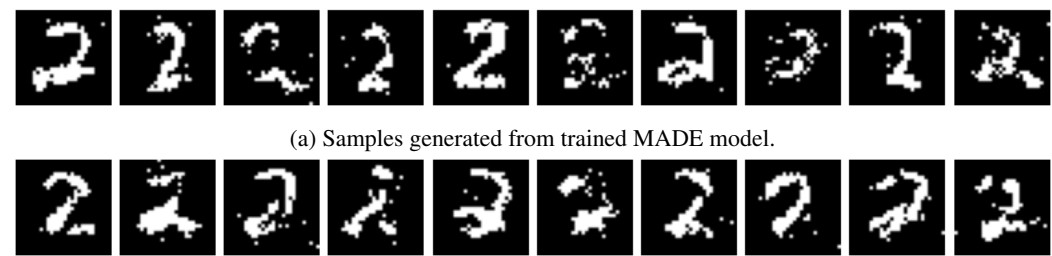

(a) Samples generated from trained MADE model.

(b) Samples generated from trained StrNN model with `nbr_size=10`.

Figure 9: Sample MNIST handwritten digits (label 2) generated by MADE and StrNN trained on 1000 data points. Both models generate samples of reasonable quality, while StrNN achieves higher likelihoods at the test samples, as illustrated in Figure 3.

# C    Synthetic Data Generation

In this paper, we used the following synthetic datasets:

1. $d = 800$ binary dataset where each variable depend on every other preceding variable ("Binary every_other").

2. $d = 50$ binary dataset where the adjacency matrix is randomly generated based on sparsity threshold ("Binary random_sparse").

3. $d = 20$ Gaussian dataset where each variable is dependent on 2 previous variables ("Gaussian prev_2").

4. $d = 20$ Gaussian dataset where the adjacency matrix is randomly generated based on a sparsity threshold ("Gaussian random_sparse").

5. $d = 15$ randomly generated non-linear and multi-modal dataset with sparse conditional dependencies between variables used for autoregressive flow evaluation.

6. $d = 5$ and $d = 10$ randomly generated datasets following linear SEMs with sparse adjacency matrices for interventional and counterfactual evaluations. The details are described in Appendix D.2.

## C.1 Adjacency Structures

In this section we visualize the adjacency matrices that were used to generate the synthetic datasets listed above. The procedure to generate each synthetic dataset given these ground truth adjacency matrices are described in the following sections. We also use these adjacency matrices to perform weight masking using StrNN/StrAF, and as the input masks for the Graphical Normalizing Flow. We visualize the adjacency matrices used by the binary experiments in Section 5.1 in Figure 10.

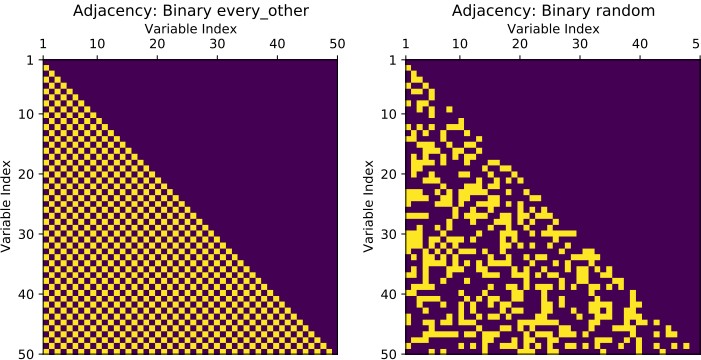

Figure 10: Adjacency matrix used to generate datasets for binary experiments. *Left:* every_other generation scheme. Note in our actual experiments, we used a 800-dimensional version of this adjacency structure. The 50-dimensional matrix is shown here for illustration only. *Right:* random_sparse generation scheme. Conditionally dependent variables are shown in yellow. These adjacency matrices are also used to generate StrNN mask matrices.

The true underlying conditional independence structure for the MNIST dataset used in Section 5.1 is unknown, which is also a common challenge for any real-world/image dataset. Instead, when using the StrNN masked neural network, we aim to encode the inductive bias of *locality*, so that density estimation for a single pixel only depends on its surrounding neighbourhood of pixels. For the results shown in the main text, we decided to use a neighbourhood size of 10 after an extensive hyperparameter search for this parameter. As explained briefly in Section 5.1, the hyperparameter `nbr_size` specifies the radius of the square context window originating from each pixel. Each pixel is modelled to be dependent on all previous pixels in that window, as specified by the variable ordering. For variable ordering, we use the default row-major pixel ordering for the MNIST images. The resulting adjacency matrix is visualized in Figure 11.

Next, we visualize the adjacency matrices that were used to generate the Gaussian synthetic datasets in Section 5.2 in Figure 12. These matrices are also used to generate the mask matrices for the StrNN.

Finally, in Figure 13 we visualize the adjacency matrix that was used to generate the non-linear multi-modal dataset in Section 5.3. This adjacency matrix is used to generate masks for StrAF, and is provided to GNF as the ground truth adjacency matrix for its input masking scheme.

## C.2 Binary Synthetic Data

We observe that based on the autoregressive assumptions, the synthetic data generating process should draw each $x_i$ as a Bernoulli random variable (i.e., a coin flip) based on $x_1, ..., x_{i-1}$, for $i = 1, ..., d$.

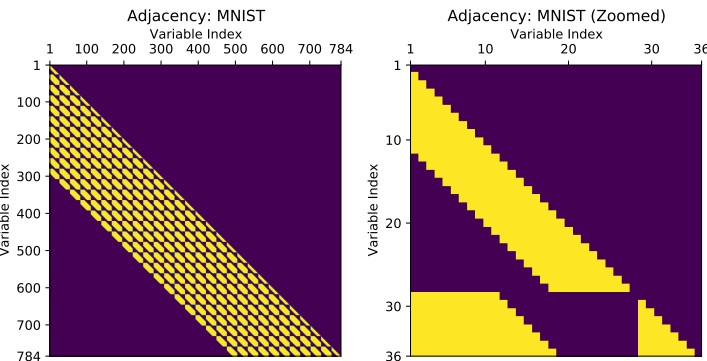

Figure 11: Adjacency matrix used to mask StrNN for the binary MNIST density estimation task, with neighbourhood size set to 10 (`nbr_size=10`). Conditionally dependent variables are shown in yellow. *Left:* All variables. *Right:* Zoomed in view of first 36 variables for illustrative purposes.

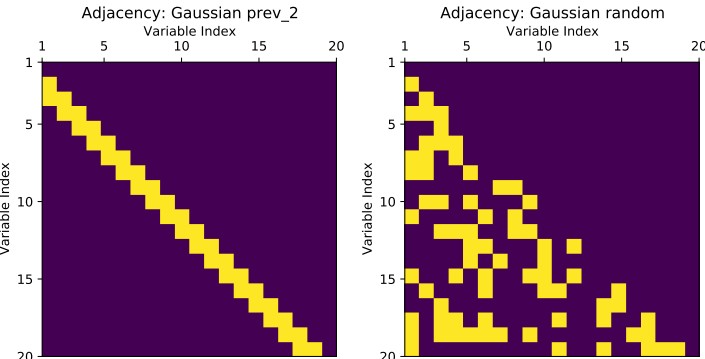

Figure 12: Adjacency matrices used to generate the Gaussian synthetic dataset. Matrices are also used to generate StrNN masks during density estimation tasks. Conditionally dependent variables are shown in yellow. *Left:* prev_2 generation scheme *Right:* random generation scheme.

Given an adjacency matrix $A \in \{0, 1\}^{d \times d}$, the general structure equations are given by:

$$x_i \sim \text{Bernoulli}(p_i), \ p_i = \text{Sigmoid}(\sum_{j=1}^{i-1} \alpha_{ij} x_j + c_i), \tag{7}$$

where $\alpha_{ij} = 0$ if $A_{ij} = 0$, otherwise $\alpha_{ij} \sim \mathcal{N}(0, 1)$ and $c_i \sim \mathcal{N}(0, 1)$.

### C.3    Gaussian Synthetic Data

Analogous to binary synthetic data generation, for Gaussian data, we sample each variable as:

$$x_i \sim \mathcal{N}(\mu_i, \sigma_i) \text{ where } \mu_i = \sum_{j=1}^{i-1} \alpha_{ij} x_j + c_i, \ \sigma_i \sim \mathcal{N}(0, 1), \tag{8}$$

where $\alpha_{ij} = 0$ if $A_{ij} = 0$, otherwise $\alpha_{ij} \sim \mathcal{N}(0, 1)$.

### C.4    Synthetic Data for Autoregressive Flow Evaluation

In this section, we describe the data generating process for the experiments in Section 5.3. The objective is to generate a $d = 15$-dimensional multi-modal and non-linear dataset. We create the ground truth adjacency matrix $A \in \{0, 1\}^{15 \times 15}$ by sampling each entry in the matrix independently from the Uniform$(0, 1)$ distribution. Each element is converted to a binary value using a sparsity threshold of $0.8$. Moreover, upper triangular elements are then zeroed out. This results in a sparse

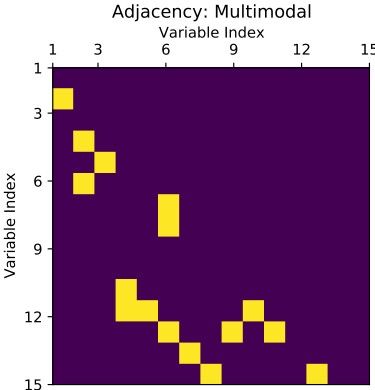

Figure 13: Adjacency matrices used to generate the multi-modal non-linear synthetic dataset used in Section 5.3. The matrix is also used to generate StrAF masks and for GNF masking. Conditionally dependent variables are shown in yellow.

binary adjacency matrix for which values of one indicate conditional dependence, and zeros indicate conditional independence. The exact matrix is visualized in Appendix C.1.

In our data generating process, variables with conditional dependencies are generated as a weighted sum of its preceding dependent variables. We generate a second matrix $W \in \mathbb{R}^{15 \times 15}$ containing these weights, where each element is sampled from the Uniform$(-3, 3)$ distribution. This matrix $W$ is then multiplied element-wise by $A$ to zero out pairs of variables that are conditionally independent. If we denote entries of $W$ as $w_{ij}$, each dependent pair of variables is generated by the following process:

$$x_t = \sqrt{\sum_{j=1}^{t-1}(w_{tj}x_j)^2} + \varepsilon, \quad \varepsilon \sim \mathcal{N}(0, 1). \tag{9}$$

Variables which are conditionally independent (e.g., $x_1$) are generated using a mixture of three Gaussians. For each variable and each Gaussian mixture component, we sample its mean from the Uniform$(-8, 8)$ distribution, and its standard deviation from the Uniform$(0.01, 2)$ distribution. We draw the mixture weights from the Dirichlet(1, 1, 1) distribution. At sampling time, we use this mixture weight vector to determine the number of samples to draw from each Gaussian mixture component. For our experiments, we draw 5000 samples using this data generating process, and use a [0.6, 0.2, 0.2] ratio for training / validation / testing splits.

# D   Causal Inference

## D.1   Algorithms

The flow-based models can be easily employed to answer causal queries, and similar to CAREFL Khemakhem et al. [2021], we provide the corresponding algorithms using flows for causal queries in this section. For the following algorithms, we denote the forward transformer $\tau$ and the flow $\mathbf{T}$ as the sampling step which transforms the noise $\mathbf{z}$ to data $\mathbf{x}$. Similarly, the backward $\tau^{-1}$ and $\mathbf{T}^{-1}$ indicates the process of taking data to noise. This notation aligns with the setup in Section 3.3.

### D.1.1   Interventions

To generate samples under intervention $do(x_i = \alpha)$, we can simply modify the corresponding structural equation of the intervened variable by breaking the connection of $x_i$ to $\mathbf{x}_{<\pi(i)}$ and setting $x_i$ to the intervened value of $\alpha$. Then, we can propagate newly sampled latent variables through the modified sampling process of the flow as shown in Algorithm 4.

If the flow only requires one pass for sampling such as inverse autoregressive flows which prevents us from changing the sampling process of each individual dimension, we can generate one random

---
**Algorithm 4:** Generate interventional samples (sequential)
---
**Data:** interventional variable $x_i$, intervention value $\alpha$, number of samples $S$
1   for $s = 1$ to $S$ do:
2      sample $\mathbf{z}(s)$ from flow base distribution (the value of $z_i$ can be discarded)
3      set $x_i(s) = \alpha$
4      for $j = 1$ to total dimension $d$ except $j = i$ do:
5          compute $x_j(x) = \tau_j(z_j(s), \mathbf{x}_{<\pi(j)}(s))$
6      end for
7   end for
8   return interventional sample $\mathbf{X} = \{\mathbf{x}(s) : s = 1, ..., S\}$
---

sample $\mathbf{x}$ first and then invert the flow to modify $z_i$ so that the corresponding $x_i$ equals to the intervened value $\alpha$. We additionally give Algorithm 5 for flows that only require one pass for sampling, such as inverse autoregressive flows.

---
**Algorithm 5:** Generate interventional samples (parallel sampling)
---
**Data:** interventional variable $x_i$, intervention value $\alpha$, number of samples $S$
1   for $s = 1$ to $S$ do:
2      sample $\mathbf{z}(s)$ from flow base distribution (the value of $z_i$ can be discarded)
3      compute initial sample through $\mathbf{x}(s) = \mathbf{T}(\mathbf{z}(s))$
4      set $z_i(s) = \tau_i^{-1}(\alpha, \mathbf{x}(s))$
5      compute final sample $\mathbf{x}(s) = \mathbf{T}(\mathbf{z}(s))$
6   end for
7   return interventional sample $\mathbf{X} = \{\mathbf{x}(s) : s = 1, ..., S\}$
---

### D.1.2   Counterfactuals

Furthermore, computing counterfactuals is typically more challenging for many causal models as it requires inferring the latent variables $\mathbf{z}$ conditioned on the observed data $\mathbf{x}_{\text{obs}}$. However, with the invertible nature of flows, it becomes straightforward to perform counterfactual inference as we have access to both forward and backward transformations between $\mathbf{x}$ and $\mathbf{z}$. Here, we present Algorithm 6 for computing counterfactuals with flows.

---
**Algorithm 6:** Compute counterfactual values
---
**Data:** observed data $\mathbf{x}_{obs}$, counterfactual variable $x_i$ and value $\alpha$
1   infer $\mathbf{z}_{obs}$ from observed data $\mathbf{z}_{obs} = \mathbf{T}^{-1}(\mathbf{x}_{obs})$ (the value of $z_i^{obs}$ can be discarded)
2   initialize $\mathbf{z}_{ctf} = \mathbf{z}_{obs}$
3   set $z_i^{ctf} = \tau_i^{-1}(\alpha, \mathbf{x}_{<\pi(i)}^{obs})$
4   compute counterfactual data $\mathbf{x}_{ctf} = \mathbf{T}(\mathbf{z}_{ctf})$
5   return $\mathbf{x}_{ctf}$
---

### D.2   Data Generation

Here, we describe the data generating process for causal inference evaluations in Section 5.4. To highlight the benefits from incorporating graphical structures, we generate both 5- and 10-variable SEMs with sparsely dependent variables, following the same procedures as outlined below.

We first generate the adjacency matrix specifying the dependencies among the variables. Each entry of the matrix is sampled from a Uniform(-2, 2) distribution. We explicitly set any entries with absolute values smaller than 1.5 along with all upper triangular entries to zero. This creates a sparse matrix with zeros representing the independencies and non-zero elements showing the strength of variable dependencies. The corresponding matrices are visualized in Figure 14.

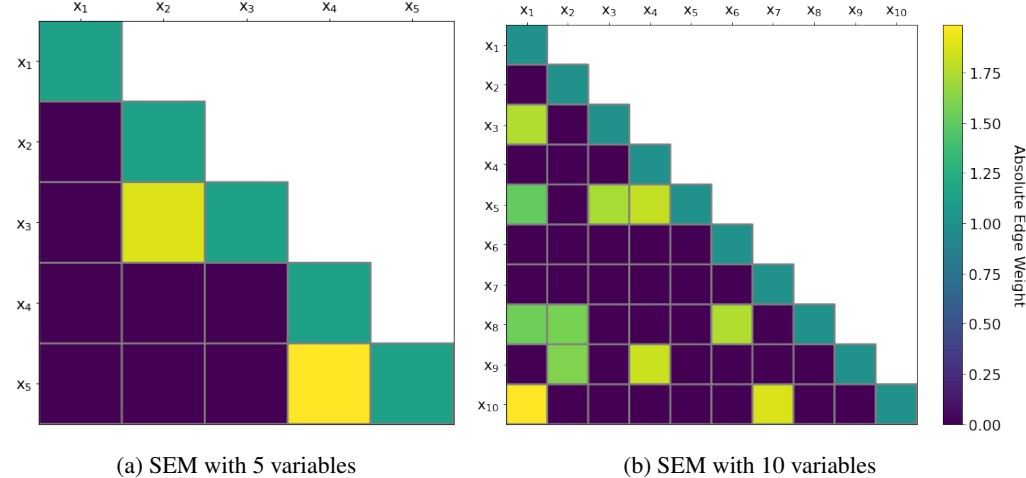

(a) SEM with 5 variables          (b) SEM with 10 variables

Figure 14: Adjacency matrices containing the coefficients relating variables in the linear additive SEM. Each row indicates the observed variable, and each column indicates the contribution of that variable to the observed variable. The absolute value of each coefficient is shown, and coefficients with zero weight indicate conditional independence.

Then, we generate synthetic data according to a linear additive structural equation model. Given an $d$-dimensional variable $\mathbf{x}$ whose distribution is defined by the SEM, each dimension of $\mathbf{x}$ follows the linear relationship:

$$x_i = \sum_{j=1}^{i-1} w_{ij} x_j + \epsilon_i. \tag{10}$$

The weights $w_{ij}$ on the preceding variables came from the generated sparse adjacency matrix, and the additive noise variable $\epsilon_j$ is sampled independently from a standard Gaussian distribution. Observed data corresponding to the SEM is generated via ancestral sampling. We first draw a sample from the noise terms and then sequentially compute the observed variables according to the relationship defined by the adjacency matrix.

### D.3 Evaluation Metrics

As mentioned in Section 5.4, we propose two evaluation metrics, total intervention mean squared error (total I-MSE) and total counterfactual mean squared error (total C-MSE), to comprehensively assess the ability of a causal generative model for answering causal queries. In this section, we present additional details, highlight key distinctions, and provide formal definitions for these metrics.

#### D.3.1 Interventions

We draw interventional samples from both the underlying true SEM where $x_i \sim \mathbb{P}(x_i|do(x_j = \alpha))$ and the causal generative flows where $\bar{x}_i \sim \mathbf{T}(x_i|do(x_j = \alpha))$. To estimate the expectation of the interventional values for the $i$-th dimension, we draw $K = 1000$ samples each for $x_i$ and $\bar{x}_i$. We can then compute the following squared error over different sample dimensions $i$, intervened variables $j$, and the corresponding value $\alpha$:

$$\text{I-Error}(i, j, \alpha) = \left( \frac{1}{K} \sum_{k=1}^{K} x_i^k - \frac{1}{K} \sum_{k=1}^{K} \bar{x}_i^k \right)^2. \tag{11}$$

We select a set $A$ of intervened values to evaluate the models. The intervened value $\alpha$ came from one of $|A| = 8$ integers perturbed around the mean of the intervened variable $x_i$. Note that we exclude interventions on all preceding variables of the intervened variable as they remain unaffected by the interventions. Hence, the total I-MSE can be obtained by averaging over all possible interventional

queries:

$$\text{Total I-MSE} = \frac{1}{|A| \times d \times (d+1)/2} \sum_{i=1}^{d} \sum_{j=1}^{i} \sum_{\alpha \in A} \text{I-Error}(i, j, \alpha), \tag{12}$$

where $d$ is the total dimension of the data $\mathbf{x}$ generated from the SEM.

### D.3.2 Counterfactuals

Similarly, we compute the counterfactual values $x_i$ from the ground truth as well as $\bar{x}_i$ from the flow models, and we use $i$, $j$, and $\alpha$ to denote the dimension to compute, the counterfactual variable, and the associated value. For counterfactuals, we compute the results based on observed data rather than generating new samples; hence, we collect $K = 1000$ observed data $\mathbf{x}_{obs}$ and pose various counterfactual queries to each observed data point. We can derive the following error averaged over different $\mathbf{x}_{obs}$:

$$\text{C-Error}(i, j, \alpha) = \frac{1}{K} \sum_{k=1}^{K} \left( x_i^k - \bar{x}_i^k \right)^2. \tag{13}$$

We also select a set $A$ of counterfactual values with $|A| = 8$ and exclude evaluations on all preceding variables of the counterfactual variable $x_j$. Then, by considering all possible counterfactual queries, the total C-MSE can be similarly written as:

$$\text{Total C-MSE} = \frac{1}{|A| \times d \times (d+1)/2} \sum_{i=1}^{d} \sum_{j=1}^{i} \sum_{\alpha \in A} \text{C-Error}(i, j, \alpha), \tag{14}$$

where $d$ again is the total dimension of the data $\mathbf{x}$ generated from the SEM.

### D.4 Experimental Setup

The causal inference experiments are relatively robust to different hyperparameter settings. We mostly follow the same setting as in CAREFL. Each model is trained using the Adam optimizer with learning rate of $0.001$ and $\beta = (0.9, 0.999)$, along with a scheduler decreasing the learning rate by a factor of $0.1$ on plateaux. We train for $750$ epochs with a batch size of $32$ data points. The flow contains 5 layers of sub-flows, and the transformers are 4-layer MADE-like networks with 10 hidden units that incorporate the corresponding autoregressive and graphical structures with masking.

## E Experiment Setup

### E.1 Experimental Setup for Binary and Gaussian Density Estimation

In this section, we describe the experimental setup for the binary and Gaussian density estimation tasks reported in Sections 5.1 and 5.2.

**Method Selection.** We compared StrNN using the greedy mask factorization algorithm (Algorithm 1) to the fully autoregressive MADE baseline as proposed in Germain et al. [2015]. MADE serves as a natural baseline since both methods use the outputs of an autoencoder to parameterize marginal probabilities, while our StrNN method has the added capability of enforcing additional conditional independence properties.

**Training.** Each model is trained with the AdamW optimizer with a batch size of 200 for a maximum of 5000 epochs.

**Hyperparameters.** We employed a grid search to find the optimal hyperparameters for StrNN and MADE respectively, where the grid is provided in Table 4. The number of hidden layers is varied during the hyperparameter search, and the number of hidden units in each hidden layer is determined by the input dimension $d$ times the hidden size multiplier of that layer. The Best hyperparameters for each model, dataset, and sample size discussed in Sections 5.1 and 5.2 are not listed here since there are too many combinations. Please refer to the code repositories for reproducing the results.

**Evaluation Metrics.** Results from binary experiments are reported in terms of the negative log-likelihood (NLL) in Figures 3 and 4. Note that in the binary case, the NLL can simply be rewritten as

the binary cross-entropy loss:

$$-\log p(\mathbf{x}) = \sum_{j=1}^{d} -\log p(x_j|\mathbf{x}_{<j}) = \sum_{j=1}^{d} -x_j \log \hat{x}_j - (1 - x_j)\log(1 - \hat{x}_j). \quad (15)$$

The results from the Gaussian experiments are also reported in NLL, which is calculated by using the neural network outputs as the parameters in each marginal conditional of the Gaussian distribution. The error ranges for the results from these experiments are computed as standard error across samples in the held-out test set.

| Hyperparameter | Grid Values |
|---|---|
| Activation | [relu] |
| Epsilon | [1, 0.01, 1e-05] |
| Hidden size multiplier 1 | [1, 4, 8, 12] |
| Hidden size multiplier 2 | [1, 4, 8, 12] |
| Hidden size multiplier 3 | [1, 4, 8, 12] |
| Hidden size multiplier 4 | [1, 4, 8, 12] |
| Hidden size multiplier 5 | [1, 4, 8, 12] |
| Number of hidden layers | [1, 2, 3, 4, 5] |
| Learning rate | [0.1, 0.05, 0.01, 0.005, 0.001] |
| Weight decay | [0.1, 0.05, 0.01, 0.005, 0.001] |

Table 4: Hyperparameter grid: StrNN vs. MADE

## E.2 Experimental Setup for Normalizing Flow Evaluations

In this section, we describe the hyperparameters and metrics used for the experimental evaluation between the Structured Autoregressive Flow (StrAF), Structured Continuous Normalizing Flow (StrCNF), and other flows in Section 5.3.

**Method Selection.** We compared the StrAF against a fully autoregressive flow (denoted ARF in the main text) and the Graphical Normalizing Flow (GNF) Wehenkel and Louppe [2021]. The fully autoregressive flow assumes no conditional independencies in the data generation process, and uses a full lower triangular adjacency matrix for masking. Meanwhile, the GNF model encodes conditional independencies, and we provide it access to the true adjacency matrix. For continuous flows, we compare the StrCNF against the FFJORD baseline and the model provided by Weilbach et al. [2020]. The FFJORD baseline uses a fully connected neural network to represent the flow dynamics, while the Weilbach et al. [2020] model can inject structure only into neural networks with a single hidden layer. This can be circumvented by stacking these neural networks, but this is shown to do worse than StrCNF in our experiments. We evaluate each model on the synthetic data described in Appendix C.4.

**Training.** Each model is trained using the Adam optimizer for a maximum of 150 epochs using a batch size of 256. During all runs, the models were trained using early stopping on the validation log-likelihood loss with a patience of 10 epochs, after which the model state at the best epoch was selected. We consider two additional training schedules: decreasing the learning rate by a factor of 0.1 on plateaus where the loss does not improve for five epochs (denoted Plateau), and a single scheduled decrease by a factor of 0.1 at epoch 40 (denoted MultiStep). In addition to the standard fixed learning rate, we select between these training schedules as a hyperparameter.

While the GNF can learn an adjacency matrix from data, we are interested in scenarios where an adjacency matrix is prescribed. Thus, we disable the learning functionality of the adjacency in GNF by stopping gradient updates to the GNF input mask matrix. We retain the one hot encoding network described in the GNF paper. The fully autoregressive flow is implemented by using a GNF with a full lower triangular adjacency matrix. Latent variables are permuted in ARF and GNF, as described by their original publications, but we do not permute variables for StrAF.

**Discrete Flow Hyperparameters.** Here we report the process used to select model hyperparameters for discrete flows. We use the UMNN [Wehenkel and Louppe, 2019] as each flow's transformer. We use 20 integration steps to compute the transformer output. The UMNN is conditioned on values computed by the conditioner. We select the dimension of these values as a hyperparameter named

| Hyperparameter | Grid Values |
|---|---|
| Flow Steps | $[1, 5, 10]$ |
| Conditioner Net Width | $[25, 50, 500, 1000]$ |
| Conditioner Net Depth | $[2, 3, 4]$ |
| UMNN Hidden Size | $[10, 25, 50]$ |
| UMNN Width | $[100, 250, 500]$ |
| UMNN Depth | $[2, 4, 6]$ |
| Learning Rate | $[0.001, 0.0001]$ |
| LR Scheduler | [Fixed, Plateau, MultiStep] |

Table 5: Hyperparameter grid for discrete flows

"UMNN Hidden Size". We then determine the best hyperparameters for each method using a grid search with the values in Table 5.

The best hyperparameters (as determined by validation loss) that were selected for each model used in the main text are reported in Table 6.

| Hyperparameter | ARF | GNF | StrAF |
|---|---|---|---|
| Flow Steps | 5 | 5 | 10 |
| Conditioner Net Width | 500 | 500 | 500 |
| Conditioner Net Depth | 4 | 2 | 2 |
| UMNN Hidden Size | 25 | 50 | 25 |
| UMNN Width | 250 | 250 | 250 |
| UMNN Depth | 6 | 6 | 6 |
| Learning Rate | 0.001 | 0.001 | 0.001 |
| LR Scheduler | Plateau | Plateau | Plateau |

Table 6: Final hyperparameters used per flow model

We fix the best hyperparameters for each model, and then re-train each model on the same data splits using eight new random seeds. This ensemble of eight models was used to generate confidence intervals for the test NLL.

**Continuous Flow Hyperparameters.** Here we report the hyperparameters used in the continuous normalizing flow models. We use the FFJORD implementation of the CNF as a baseline, and then substitute the neural network representing dyanmics in the architecture with the Weilbach et al. [2020] masked neural network, or the StrNN. For the ODE solver, we generally use the default FFJORD hyperparameters, as reported in Table 7.

| Hyperparameter | Values |
|---|---|
| ODE-NN Activation Function | Tanh |
| ODE Time Length | 0.5 |
| ODE Train Time | True |
| ODE Solver | dopri5 |
| ODE Solver Absolute Tolerance | 1e-5 |
| ODE Solver Relative Tolerance | 1e-5 |

Table 7: Fixed hyperparameters for CNF models.

Other hyperparameters were selected using validation loss from the following grid in Table 8. We again note that the neural network representing ODE dynamics for the Weilbach model is adapted to have multiple hidden layers by stacking multiple neural networks that have a single hidden layer. The final hyperparameters for each model are reported in Table 9. As before, the final test NLL results in the main text are reported after fixing hyperparameters, and then training 8 models from random initialization.

| Hyperparameter | Grid |
|---|---|
| Flow Steps | [1, 5, 10] |
| ODE-NN Width | [50, 500] |
| ODE-NN Depth | [2, 3, 8] |
| Learning Rate | [5e-3, 1e-3, 1e-4] |

| Hyperparameter | FFJORD | Weilbach | StrCNF |
|---|---|---|---|
| Flow Steps | 5 | 10 | 10 |
| ODE-NN Width | 50 | 500 | 500 |
| ODE-NN Depth | 2 | 3 | 2 |
| Learning Rate | 5e-3 | 5e-3 | 5e-3 |

Table 8: Hyperparameter grid for CNF models.     Table 9: Final hyperparameters for CNF models.

## E.3 Graphical Normalizing Flow Baseline

We use the official implementation of Graphical Normalizing Flows (GNF) as provided in the publication Wehenkel and Louppe [2021]. In our experiments, we found issues when using the code to evaluate the reverse flow, i.e., the transformation from the latent space to the data distribution. The code we believe to be problematic has been copied below with minor edits for clarity. The original version can be found at: `https://github.com/AWehenkel/Graphical-Normalizing-Flows/blob/2cc6fba392897ec1884b4f01a695b83d3c04883a/models/NormalizingFlow.py#L166`.

```python
def forward(self, x):
    inv_idx = torch.arange(x.shape[1] - 1, -1, -1).long()
    for step in self.steps:
        z = step(x)
        x = z[:, inv_idx]
    return z

def invert(self, z):
    for step in range(len(self.steps)):
        z = self.steps[-step].invert(z)
    return z
```

Typically, if we denote the forward flow transformation as $\mathbf{T} = \mathbf{T}_1 \circ ... \circ \mathbf{T}_K$, then the reverse flow should be computed as $\mathbf{T}^{-1} = \mathbf{T}_K^{-1} \circ ... \circ \mathbf{T}_1^{-1}$. The GNF code implemented in Python, however, computes the reverse flow as $\mathbf{T}^{-1} = \mathbf{T}_1^{-1} \circ \mathbf{T}_K^{-1} \circ \mathbf{T}_{K-1}^{-1} ... \circ \mathbf{T}_2^{-1}$. These reverse flow transformations are only the same when $\mathbf{T}_1 = \mathbf{Id}$, but in general there is no guarantee they are equivalent when learning all layers in the flow, i.e., $\mathbf{T}_1 \neq \mathbf{Id}$.

More importantly, while the forward transformation permutes latent variables between flow steps, the inverse transformation does not perform the same permutation. We believe that this choice harms sample quality generation, as can be seen in our experimental section. However, we could not find reference to the inversion process in the GNF publication, and thus were unable to determine if this was a bug in their code, or an actual limitation of the method. As such, we decided to simply use the official GNF code to produce the results for our evaluations.