# OpenReview forum: "Structured Neural Networks for Density Estimation and Causal Inference"
_NeurIPS.cc/2023/Conference — NeurIPS 2023 poster_

### Official Review · Reviewer_LoZH · 2023-06-26

**Soundness:** 3 good
**Presentation:** 3 good
**Contribution:** 2 fair
**Rating:** 6
**Confidence:** 3

**Summary:**

This work studies the impact of imposing known conditional independence structure in fully-connected neural network architectures, notably in the setting of autoregressive normalizing flows. The independence is imposed by masking the weight matrices in the linear layers, similar to the MADE approach. The masks are determined by a factorization of the known adjacency matrix that (approximately) maximizes the number of connections (paths) between inputs and outputs. Through several experiments, the authors show generalization improvement over baselines.

**Edit** Most of my concerns were addressed during the rebuttal/discussion period. I am therefore upgrading my score from 4 to 6.

**Strengths:**

* The manuscript is well written, with standard notations.
* The contributions are clearly defined and the assumptions (known adjacency) are explicit.
* The proposed approach to impose conditional independences is sound.
* The claims are supported by the experiments, notably concerning improved generalization.

**Weaknesses:**

The idea of imposing prior independence knowledge into autoregressive flows was first introduced by Wehenkel and Louppe (2021), cited as [25] in this manuscript. As the authors mention (lines 246-247), StrAF only differs from GNF in the approach to impose the independences, but is conceptually identical. Actually, Wehenkel and Louppe (2021) already propose the approach of the present work:

    > An alternative approach would be to use masking scheme similar to what is done by Germain et al. (2015) in MADE as suggested by Lachapelle et al. (2019).

In addition, the official [UMNN repository](https://github.com/AWehenkel/UMNN), also cited in this work, links to the normalizing flow library [Zuko](https://github.com/francois-rozet/zuko), from the same lab. The latter library implements autoregressive flow conditioners as masked multi-layer perceptrons for which the masks are a factorization of the adjacency matrix between the inputs and outputs. The similarities with the proposed StrNN are too strong to be left unaddressed.

**Questions:**

* Algorithm 1: It is not clear to me how the factorization algorithm is applied when the StrNN has more than one hidden layer.
* Section 5.1: Are the same number of layers/neurons used for StrNN and MADE in this experiment?
* Line 317: "As GNF permutes variables between flow steps". I was not able to find a mention of this in [25].
* Figure 5/Table 1: I don't understand how "ARF-10" and "GNF-10" can be so much worse than "GNF-1", as they are strictly more expressive. Is it an overfitting issue? Or maybe an invertibility issue? UMNN is not always numerically invertible. What about "ARF-1"?
* Table 1: The authors make a distinction between "density estimation" and "sample quality", which does not make sense to me. If a flow perfectly estimates the density, it necessarily generates probable samples, unless the invertibility is not guaranteed.
* Why not studying the use of StrNN in other settings than density evaluation, such as physics-informed machine learning, where it is common to infuse prior knowledge in the structure of the neural networks?

**Limitations:**

* It is never mentioned that a StrNN is a (pruned) fully-connected network with element-wise activation functions. The approach does not apply to convolutional, attention-based or recurrent networks, and does not support skip/residual connections or normalization layers.
* This is not a limitation of this work, but one should be careful not to confuse Bayesian networks and causal graphs. A Bayesian network (or its adjacency matrix) over variables merely indicates independencies between the variables but in no way causalities.

---

> ### Author Rebuttal · Authors · 2023-08-09
>
> We thank reviewer LoZH for their detailed feedback, especially for highlighting the clarity and soundness of our approach to enforce conditional independences in NNs.
>
> We begin with the Zuko repo, which is relevant to our work. First, are there manuscripts or research results using Zuko that we could cite? We could not find any based on the GitHub authors' profiles. Otherwise we will reference the repo in our revision.
>
> Second, we note that the **code in the Zuko repo (as cloned on the NeurIPS submissions deadline)** **does not allow users to initialize any of its flows to encode a global adjacency structure**, and attempting to do so gives an error.
>
> Third, **while Zuko and StrNN’s greedy factorizations of the adjacency matrix are similar, they yield different results.** In the experiment below, mask matrices found by our greedy algorithm and by Zuko result in different performance. Zuko operates on the unique rows of an adjacency matrix, which causes issues when the matrix contains many repeating rows. This type of structure may be naturally encountered in datasets with star shaped graphs. Consider a NxN matrix where the first (N-1) rows contain dependence on the first variable ([1, 0, …, 0]) and only the last row depends on the second variable ([0, 1, 0, …, 0]). Given a budget of H hidden units, Zuko would assign H // 2 units to represent the variable corresponding to the last row. This ineffectively represents the dependence of all other outputs on the first variable, and is avoided by our greedy algorithm. In a non-linear dataset with N variables and the above adjacency matrix, we compare StrAF with our greedy algorithm and Zuko’s algorithm in the table below. We report runs from 5 random seeds, using a 95% CI and N hidden units.
>
> | Model | Test NLL (N=50) | Test NLL (N=1024) |
> | --- | --- | --- |
> | StrNN-Greedy | 0.4311 ± 0.5406 | -18.734 ± 1.999 |
> | Zuko | 1.3276 ± 0.2806 | -14.550 ± 0.936 |
>
> We see our method performs better, especially as N increases. **This supports our claim that the choice of mask factorization can impact performance beyond enforcing independences.**
>
> Fourth, Wehenkel and Louppe (2021) and Lachapelle et al (2019) did suggest a masking scheme like MADE. **But there is a gap between the suggestion of an idea and a practical instantiation thereof.** Neither works studied any concrete algorithms or experiments to this end, and only raised the possibility of such an approach for future work. **We believe our work establishes the feasibility of the method, conducts rigorous evaluations, and demonstrates its application in a novel context.**
>
> We reply to other questions and concerns below:
>
> > Algorithm 1: … how the … algorithm is applied when ... StrNN has more than one hidden layer.
> >
>
> **We apply the factorization algorithm recursively, layer-by-layer, for efficiency and simplicity.** Given $A\in \{0, 1\}^{d \times d}$, we run Algorithm 1 once to find $ A_1 \in \{0, 1\}^{d \times h_1}$ and $ M^1 \in \{0, 1\}^{h_1 \times d}$ for a layer of width $h_1$ such that $A_1\cdot M^1 \sim A$, where $\sim$ denotes the matrices share the same sparsity. For the next layer with $h_2$ units, we use $A_1$ in place of $A$ to find $A_2 \in \{0, 1\}^{d \times h_2}$ and $M^2 \in \{0, 1\}^{h_2 \times h_1}$ such that $A_2 \cdot M^2 \sim A_1$. Repeat until we have all the masks.
>
> > Section 5.1: same number of layers/neurons used for StrNN and MADE …?
> >
>
> **Yes**, the NNs in StrNN and MADE have the same number of layers/hidden units to control for capacity.
>
> > Line 317: "As GNF permutes variables …". … not able to find a mention of this in [25].
> >
>
> **See the following link (with line #) in the GNF repo where the authors permute variables between flow steps:** https://tinyurl.com/vs92kncc. Variable permutation is common in the normalizing flow literature. We believe pointing out the necessity of properly handling adjacency masks with respect to this operation to be a simple yet impactful observation.
>
> > Figure 5/Table 1:  how "ARF-10" and "GNF-10" can be so much worse than "GNF-1" …
> >
>
> **High expressivity of a model does not necessarily lead to better generalization.** Though we have conducted hyper-parameter/model selection for all methods, as detailed in Appendix E.2, the baseline models can still over-fit on spurious correlations. When an adjacency matrix is known, we can avoid this, leading to the improvement of GNF-1 over ARF-10. As shown in Figure 2 and the link above, GNF does not properly handle variable permutations and adjacency masking, leading to an incorrect global adjacency matrix, hence the drop in performance between GNF-1 and GNF-10. We will edit our final revision for clarity on this matter.
>
> > Table 1: … distinction between "density estimation" and "sample quality"…
> >
>
> We thank the reviewer for allowing us to clarify this subtle point. This distinction arises from the metric used to assess how *well* the flow estimates the target density. Common divergences between probability distributions used to learn the flow, e.g. KL divergence, are weighted by the target distribution. Thus, the loss is not very sensitive to the model’s performance in low-probability regions of the target. This behaviour might lead to poor “sample quality” even if the likelihood values on the test set is high.
>
> > … StrNN in other settings than density evaluation, such as physics-informed machine learning …
> >
>
> This is a great direction for future research! In our paper, we studied causal effect estimation as a novel application area. **We will keep physics-informed ML in mind for future work.**
>
> > (StrNN) does not apply to convolutional, attention-based or recurrent networks, .. does not support skip/residual connections or normalization layers.
> >
>
> **StrNN can be extended to several of these architectures.** While our focus is on MLPs, the StrNN can be applied in the linear layers of recurrent networks to enforce structure throughout the sequence. **This is beyond the scope of this work.**

---

> > ### Comment · Reviewer_LoZH · 2023-08-11
> >
> > Thank you for your answers and the experiments you have conducted in such a short time. You have addressed most of my concerns.
> >
> > I appreciate your honesty regarding the similarities between your factorization algorithm(s) and the one present in Zuko. I also appreciate you taking the time to understand and highlight the differences, which, as you demonstrate, can impact the performances beyond enforcing the independences. I would like to mention that, to my understanding, one factorization is not necessarily better than the other. They are different and will lead to different results for different adjacency structures.
> >
> > Nevertheless, you rightfully observe that, even though the factorization algorithm is present in Zuko, users are not easily able to initialize their flow to respect a particular adjacency matrix. After some research, I did not find a publication linked to the Zuko library, or to the factorization algorithm therein. I believe a reference to the repo as well as a short discussion of the similarities/differences would be enough. Finally, I agree with you that there is a gap between the suggestion of an idea and its implementation, and your work goes beyond this idea by demonstrating the impact of the factorization in terms of generalization.
> >
> > Concerning my questions, I thank you for your clear answers. I would suggest you to add some of these within the manuscript/discussion.
> >
> > In view of the rebuttal and the proposed changes, I am willing to reconsider my score. I believe your work, although light from a theory perspective, is a valuable contribution to the density estimation community.

---

> > > ### Author Response · Authors · 2023-08-11
> > >
> > > Dear Reviewer LoZH:
> > >
> > > Thank you for the prompt response to our rebuttal.
> > >
> > > We are very glad that we have managed to address most of your concerns.
> > >
> > > We agree it is unlikely there is one factorization scheme which is optimal for all possible adjacency matrices, which is a limitation we will discuss in our revision. However, we do believe some factorization objectives/algorithms are better than others on average due to how they dictate the resulting masked NN architecture. We have identified this as an important area to formalize and investigate in future work.
> > >
> > > In our rebuttal, we used one example where our greedy algorithm outperforms Zuko to highlight one key difference. At the same time, we acknowledge that there might be specific adjacency structures for which our greedy algorithm would do less well. Nonetheless, StrNN also gives us the ability to choose our factorization objective based on prior information, which we believe will help shed light on this in the future.
> > >
> > > We are currently editing our manuscript for clarity based on all reviewer suggestions and questions. Thank you again for helping us improve our work!
> > >
> > > If you do find our responses satisfactory, we would greatly appreciate it if you could re-evaluate our paper for a higher score.
> > >
> > > Thank you for your time.
> > >
> > > Best,
> > >
> > > Authors

---

### Official Review · Reviewer_icwN · 2023-06-29

**Soundness:** 3 good
**Presentation:** 2 fair
**Contribution:** 3 good
**Rating:** 6
**Confidence:** 4

**Summary:**

The authors propose a novel method for constructing structured neural networks (strNN) that are able to respect causal independencies between variables. Formal constraints for weight mask creation are discussed and evaluated empirically for an exact method and a greedy algorithm. The conditioned strNN are furthermore leveraged for conditioning autoregressive flow models. Practical application successfully is demonstrated over multiple synthetic datasets with comparisons between several baseline models, with and without the use of adjacency information. To the best of my knowledge related work is discussed sufficiently in the context of causal density estimation.

**Strengths:**

1. The authors propose a novel method for constructing structured models via weight masking that integrates seamlessly with existing neural architectures while respecting the strict independence assumptions of causal models. Preconditions and assumptions for application of the approach, specifically knowledge about the causal graph structure, are clearly stated.

2. To tackle the infeasibility of exact mask creation on larger graphs a greedy algorithm is proposed and its practical application is demonstrated.

3. The authors additionally include experiments on binary MNIST data, for which the underlying causal structure is unknown. By imposing a causal graph structure which promotes the usage of spatially local information, the authors improve performance for non-synthetic data over baselines.

**Weaknesses:**

The example shown in Figure 1 decomposes the network into two separate networks. However, constructing the displayed a network would not require a complicated mask decomposition, but could be trivially solved by constructing two independent networks with constrained layer width. Only by inspecting the example provided in the appendix it is revealed that the presence of split-structures leads to shared weights between the outputs.

**Questions:**

1. As causal mechanisms are often assumed to be independent in causal literature, I would like to ask the authors about the benefits or downsides of allowing for such shared weights within the network.

2. Furthermore, I would like to ask the authors to discuss possible simplifications for specific causal structures, e.g. in the case of independent causal mechanisms as seen in Figure 1.

Overall the idea is pretty good with a clever way of enforcing the causal independencies. However, all of this is assuming that the networks can leverage shared information between different mechanisms. If that is not the case then you could just train an independent density estimator for every single edge and (from a purely causal perspective) the problem becomes trivial to solve.

**Limitations:**

No concerns here

---

> ### Author Rebuttal · Authors · 2023-08-09
>
> We thank reviewer icwN kindly for their encouraging and constructive feedback, especially for highlighting the novelty and efficacy of our matrix factorization approach to enforcing exact independence structures in arbitrary NNs.
>
> > The example shown in Figure 1 decomposes the network into two separate networks.
> >
>
> We will use an alternative example adjacency matrix in Figure 1 which requires shared weights in our final revision. We thank the reviewer for their attention to detail on this matter.
>
> A main point in the reviewer’s feedback is regarding our design choice to use one single neural network with shared weights in StrNN. In order to enforce independence structure among variables in a single NN, we are required to solve the binary matrix factorization problem central to our paper. The reviewer is correct in saying we can instead find a separate NN for each output variable, which only takes that variable’s dependencies as inputs. That approach would be very simple and render masking and matrix factorization procedures unnecessary. However, W**e believe that the particular way we share weights in StrNN has several advantages**:
>
> 1. **Sharing weights is more efficient for improving both memory and computational efficiency against other approaches for masking.** For example, when there are D input features, using D independent networks or the GNF baseline (which masks input variables) necessarily requires D forward passes for a single input example. In comparison, we can do the same computation in a single forward pass. This becomes especially relevant in contexts such as continuous normalizing flows (CNFs), where many network evaluations must be done in the ODE solver. Here, methods with poor asymptotic complexity become impractical. In our additional results for integrating StrNNs into CNFs with the dataset from Section 5.3 (as described in the **general author response**), we find that the StrNN model outperforms a baseline FFJORD even when using a number of hidden units on the same order of magnitude (as opposed to D times more,  see **supplementary rebuttal PDF** for exact experiment details). This would not be possible unless the model leveraged weight sharing in StrNNs,  providing further empirical justification for the utility of this choice.
> 2. **From a causality perspective, one possible benefit from sharing weights could come from robustness to model misspecification. T**he best model depends on the actual problem at hand, its associated data, and possibly the true generative process. To our knowledge, an empirical study comparing sharing weights vs independent models does not exist and would be interesting for the community. Further investigation into this question using the StrAF extension of CAREFL we built in Section 5.4 of our paper would be a promising direction for future research. However, as suggested in [1], if the functional forms of the independent mechanisms are misspecified, then variables outside of the Markov blanket may improve the predictive model performance. By sharing weights inside the network, gradient information from other variables’ relationship can improve generalization of a given conditional distribution.
>
> Lastly, the reviewer asked us to discuss
>
> > possible simplification of specific causal structures, i.e., independent causal mechanisms as in Figure 1.
> >
>
> **We believe this is a fruitful research direction of applying our strNN framework to other statistical models in causal inference that may inherit sparse structure.** Two possible simplifications are:
>
> 1. **Additive noise models with a common noise mechanism**, i.e., the data-generating process for each variable $x_k$ is given by $x_k = f_k(x_1,\dots,x_{k-1}) + eps$, where eps is independent of k. In this case, the strNN outputs will only need predict the conditional expectation $f_k = \mathbb{E}[x_k | x_1,\dots,x_{k-1}]$ for all k, and a separate network is used to model the distribution of the noise variable $eps$.
> 2. **Graphical models with sub-groups of connected components**. For example, Figure 1 shows a model with four variables where (x_1,x_2) is a connected group, and (x_3,x_4) is connected to the first group by an edge from x_2. In this case, we could apply strNN to groups of variables in a hierarchical way: first to the sub-group using a single edge between then, and then to the variables within each group.
>
> Finally, we would like to extend our thanks once again to Reviewer icwN for the insightful feedback provided. We believe that the changes made to our manuscript can address your questions and concerns. We hope you will find the revisions satisfactory and consider a reevaluation of our paper's score. Should you have any further feedback or questions, please feel free to share.
>
> [1] Peters J, Janzing D, Schölkopf B. Elements of causal inference: foundations and learning algorithms. The MIT Press; 2017. Chapter 6. Multivariate Causal Models. p.103.

---

> > ### Comment · Reviewer_icwN · 2023-08-11
> > **Thank you for the Response**
> >
> > I would like to thank the authors for their response. I am happy with their response and am still positive about the paper and it's contributions.

---

### Official Review · Reviewer_Wq9i · 2023-07-06

**Soundness:** 4 excellent
**Presentation:** 3 good
**Contribution:** 3 good
**Rating:** 6
**Confidence:** 4

**Summary:**

In this paper, the authors present a neural network architecture that can fulfill the bayesian DAG conditional independencies needed for normalized density estimation.
In this work, the authors start from a binary lower adjacency matrix that encodes the independencies of a bayesian network DAG. Then they introduce a factorization of the global adjacency matrix into L adjacency matrices, which can be used as masks on each layer. This construction allows neural networks to be trained for a normalized density estimation task.
The authors introduce an heuristic to exactly factorize the adjacency matrix for the different layers using two objective functions.
With this building block the authors proceed to create a normalizing flow architecture that respects the independency restrictions end-to-end.  The authors then compare their approach empirically on different task against MADE, a neural density estimator that allows general dependencies for a given random variable ordering.

**Strengths:**

I found the paper insightful, and the results show that the approach is beneficial. The paper is technically sound and easy to read. The results showing an improvement in data efficiency for a given negative likelihood are also very interesting.

The introduction of the normalizing flow approach and the comparison in the causal setting are also nice additions that can have impact in the broader community.

The experiment on the sample quality shows the benefit of restricting the dependencies in the network as it cuts paths for noise in other random variables (and feature transformations) to propagate through the network.

**Weaknesses:**

The major weakness of this paper is the limited empirical section in comparison to other papers in this domain. This can cause readers to wonder if the benefits of the new approach as density estimators are not significant for other datasets.
A broader comparison on other datasets would make the paper more robust.

Also and I'm considering this as a minor weakness in my review, is that the method although insightful does not provide a way to obtain the global adjacency matrix.



**Questions:**

As I was reading the paper, my first thought would be that you would explore the possibility of discovering the dependencies for a given order.
Here one can start with a dense adjacency matrix, train the network, clip nodes in the layers according to Lottery Ticket Hypothesis, and propagate the masks forward, i.e., multiplying all the masks to get the adjacency matrix. As you are clipping, the adjacency matrix is guaranteed to either be the same or introduce independencies.
At that point, you can even use your factorization algorithm again to obtain a network with more/other nodes active while still respecting the new independencies.
I'm wondering if you explored similar ideas during your research?


**Limitations:**

The main limitations of the approaches presented are inherited from the restrictions on ordered models, e.g., marginalization and map queries are intractable for the general case. This is not mentioned in the paper.

There are no potential negative societal impact to this work.

---

> ### Author Rebuttal · Authors · 2023-08-09
>
> We would like to warmly thank Reviewer Wq9i for their encouraging feedback, especially for highlighting our method’s emphasis on data efficiency and the novelty of the causal effect estimation application.
>
> The reviewer raises a good point on how broader comparison on other datasets would make our paper more robust. While the synthetic datasets included in our evaluation are highly non-linear and challenging to model, we are working on extending our method to several real world datasets. We have obtained favorable density estimation results in adding StrNN to continuous normalizing flows such as FFJORD, which we have included in the general author response. This further validates StrNN’s efficacy as a density estimation technique.
>
> **re: StrNNs and structure discovery**
>
> > Explore the possibility of discovering the dependencies for a given order. Here one can start with a dense adjacency matrix, train the network, clip nodes in the layers according to Lottery Ticket Hypothesis, and propagate the masks forward … I'm wondering if you explored similar ideas during your research?
> >
>
> That is a great point! Overall, structure discovery is outside the scope of our paper since we assume either the ground truth Bayesian network or a good estimate is known. However, **we are interested in pursuing future work in using StrNN to learn structure from data, thus providing a full pipeline from structure discovery to density estimation and sample generation.**
>
> **We have considered the following potential connections to dropout and the lottery ticket hypothesis during our work.**
>
> First, we noticed that stochastically introducing sparsity into a neural network has interesting connections to dropout, which is also known to improve model generalization. It would be interesting to investigate if specific patterns of sparsity emerge from this stochastic process, and as the reviewer points out, if our method provides a framework with which to directly optimize for any specific qualities of sparsity while respecting structure.
>
> Second, we have considered a gradient-based approach to learn structure from data as well, where we might add a group regularization penalty to the weights in order to penalize the dependence of certain marginal conditionals on individual input variables.
>
> > The main limitations of the approaches presented are inherited from the restrictions on ordered models, e.g., marginalization and map queries are intractable for the general case. This is not mentioned in the paper.
> >
>
> Lastly, we thank the reviewer for raising an important point on the core limitation of all autoregressive models, which is the difficulty of performing marginal and MAP inference. We will make this point clearer in our final revision.
>
> We would like to thank Reviewer Wq9i once again for your constructive feedback. We are confident our revised manuscript can address your questions and concerns, and we kindly invite you to reconsider the score for our paper if you are satisfied with our responses. Please don’t hesitate to let us know of any additional comments!

---

> > ### Comment · Reviewer_Wq9i · 2023-08-14
> >
> > I thank the authors for their rebuttal. I still consider the paper in a positive light and after following the discussion, I will keep my score as is.

---

### Official Review · Reviewer_Tj4q · 2023-07-27

**Soundness:** 3 good
**Presentation:** 3 good
**Contribution:** 3 good
**Rating:** 5
**Confidence:** 4

**Summary:**

This paper introduces structured neural networks such that the resulting neural network represents the factorization of a given Bayesian network. For doing so, each layer of the neural network is masked and the product of the masks of all layers must be the same as the adjacency matrix of the DAG representing the Bayesian network.
With this construction, the represented conditional dependencies with structured neural networks will be consistent with the given Bayesian network. The paper proposes a simple greedy algorithm to find the masks. It also proposes using the structured neural network to construct the coupling layers for normalizing flow and claims that the resulting generative model is better for casual inference (intervention and counterfactuals) than the prior approach.

**Strengths:**

The paper is well-written and the contribution towards causal inference is solid.

**Weaknesses:**

1- The structured neural network augments MADE with a better mask construction algorithm such that the factorization can be defined for any DAG structure. However, it is not a fundamentally different model.
2- The paper didn't propose any approach for learning the structure of DAG given the provided structured neural network parametrization.
3- MADE is not a strong density estimator and comparing only to MADE does not validate the strength of structured neural networks as density estimators.


**Questions:**

1) How GNF would compare to StrAF if it does permute the latent variables after the first step?
2) During the comparison with CAREFL did you provide CAREFL with external DAG orders that StrAF uses? If not, the learned causal order by CAREFL may not exactly specify the underlying DAG, which may result in lower performance in causal inference.

---

> ### Author Rebuttal · Authors · 2023-08-09
>
> We would like to thank reviewer Tj4q for their feedback and insightful comments, especially for highlighting the simplicity of our matrix factorization approach and our contribution to the causal inference application.
>
> First, we hope to provide some further clarification on the questions posed by the reviewer:
>
> > How GNF would compare to StrAF if it does permute the latent variables after the first step?
> >
>
> **We believe GNF performs worse than StrAF when the latent variables are permuted after the first step.** The GNF already permutes latent variables after the first step, but does not address the effect of this operation on the conditional independencies enforced by the overall network (i.e., the resulting adjacency matrix between inputs and outputs). This is consistent with how the GNF authors have implemented their official code-base (see https://tinyurl.com/vs92kncc). As we visualize in Figure 2, this permutation can encode incorrect connectivity between the variables. Our experiments show this causes higher MMD values between samples from GNF and the true distribution, in comparison to StrAF.
>
> > During the comparison with CAREFL did you provide CAREFL with external DAG orders that StrAF uses? …
> >
>
> **Yes, we provide the ground truth DAGs to both StrAF and CAREFL during the evaluations for causal inference.** However, CAREFL can only utilize the autoregressive order provided whereas StrAF enforces the entire adjacency structure, which results in improved performance. Even though CAREFL discussed approaches for extending their pairwise discovery method to multivariate data, they assumed the autoregressive orders are given in their causal inference experiments. Similarly, one major assumption in our experiments is that the true structure has been resolved before modelling the generative process.
>
> Next, we address some of the weaknesses of our paper as identified by the reviewer.
>
> > The structured neural network augments MADE with a better mask construction algorithm such that the factorization can be defined for any DAG structure. However, it is not a fundamentally different model.
> >
>
> We believe that, **while StrNN is similar to MADE, the augmentation itself makes a significant  contribution to the literature** for the following reasons:
>
> 1. While MADE identifies an approach to enforce autoregressive dependence in NN outputs by masking weights, their algorithm does not consider sparse autoregressive dependence, which appears in many applications such as causal inference.
> 2. The way we enforce structure through matrix factorization allows us to directly specify a neural architecture. As far as we know, this important link between masking and NN architecture has not been explicitly discussed or explored in MADE or other prior works. We refer reviewer Tj4q to our overall author response for additional experiments on how the factorization objectives can impact NN performance.
>
> > The paper didn't propose any approach for learning the structure of DAG given the provided structured neural network parametrization.
> >
>
> **In the conclusions and limitations section of our manuscript, we mention that while we assume access to the true adjacency matrix, one can use any SOTA structure discovery algorithm, such as NO TEARS, in conjunction with StrNN and StrAF, as long as one keeps the flow layers un-permuted.** We will also add an additional reference to the causal-learn [1] package, which provides a variety of out-of-the-box structure discovery algorithms that can be easily integrated into our StrNN/StrAF code.
>
> **While structure discovery is outside the scope of this paper, we are interested in pursuing future work in using StrNN to learn structure from data, providing a full pipeline from structure discovery to density estimation and sample generation.** One option is to add a group regularization penalty to the network masks or weights in order to penalize the dependence of certain marginal conditionals on individual input variables. We have also thought of an alternative stochastic approach to structure discovery, by using dropout to reduce the total number of connections, which is similar to the proposal from reviewer Wq9i based on the lottery ticket hypothesis.
>
> > MADE is not a strong density estimator and comparing only to MADE does not validate the strength of structured neural networks as density estimators.
> >
>
> We compare to MADE as a baseline because it is the natural choice that assumes a full autoregressive factorization. We do not claim that StrNN is the strongest binary density estimator, rather we show that **adding structure to several existing methods improves their performance.**
>
> We later show that incorporating StrNN into popular continuous-valued density estimation methods based on autoregressive flows improves their performance. Our baseline, GNF, is a very strong density estimator, and Table 1 in our paper shows that StrAF can improve upon GNF for both test likelihood values and sample quality.
>
> To reinforce this point, we incorporated StrNN with continuous normalizing flows, FFJORD. We report results in the overall author response. We observe that adding StrNN improved performance against the baseline FFJORD model, again demonstrating the strength of StrNN, not as an independent density estimator, but as a drop-in replacement to improve the performance of density estimators by leveraging adjacency structure. We aim to further study StrNN to improve current SOTA density estimators, such as diffusion models.
>
> Lastly, we thank Reviewer Tj4q once again for your constructive feedback. We are confident our revised manuscript can address your questions and concerns, and we kindly invite you to reconsider the score for our paper if you are satisfied with our responses.
>
> [1] Zheng Y, Huang B, Chen W, Ramsey J, Gong M, Cai R, Shimizu S, Spirtes P, Zhang K. Causal-learn: Causal Discovery in Python. arXiv preprint arXiv:2307.16405. 2023 Jul 31.

---

### Author Rebuttal · Authors · 2023-08-09

We thank all reviewers for your engagement and thoughtful comments. We have addressed specific concerns and questions in individual rebuttals, but here we **highlight two experiments added based on the feedback**:

1. We extend Section 5.3 with experiments on **continuous normalizing flows (CNF)**. We choose FFJORD [1] (denote as **FFJORD-FC**), a popular strong density estimator, as our baseline, and improve it by injecting prior known structure via StrNN. In particular, we replace the feed-forward network used to model the flow dynamics with a StrNN, denoted **FFJORD-StrNN**. We also compare against [2], which also attempts to encode structure into the ODEnets of a FFJORD model (denote as **FFJORD-Weilbach**). However, FFJORD-Weilbach does not use mask factorization and directly multiplies an adjacency matrix onto a single hidden layer. Thus, they are restricted to a single hidden layer with same number of units as the input dimension, causing underperformance. We note that [2] proposes interesting orthogonal avenues of improving CNFs which we can explore in the future. Using the dataset from Section 5.3, we perform a hyperparameter search (**Table 1** in PDF document). We report Test NLL with a 95% CI based on 8 runs from random initializations below:


    | Model | Test NLL |
    | --- | --- |
    | FFJORD-FC | -3.9718 ± 0.0677 |
    | FFJORD-StrNN | **-4.0583 ± 0.0089** |
    | FFJORD-Weilbach | -1.0624 ± 0.0724 |

    Integration of StrNN results in significantly better performance against both baselines. **We believe this reinforces the comments identifying StrNN as a generally applicable, drop-in tool to improve existing density estimators.**

2. We highlight the importance of the StrNN capability to **specify different objectives in the adjacency matrix factorization algorithm**. In particular, we wish to demonstrate the consequences of sub-optimal factorizations. We show density estimation performance using two alternative factorizations: (1) **mincon**, which minimizes the number of connections while remaining faithful to the adjacency matrix, and (2) **random**, which recreates the random factorization scheme from [3]. We compare these two against our **greedy** algorithm when initializing StrNNs in a StrAF model. Our dataset is based on a 10-dimensional sparse linear additive SEM, similar to Section 5.4. We use a 5-step flow with 3 hidden layers each with 20 hidden units. We report test NLL with a 95% CI based on 5 runs from different random initializations. We also report the fraction of remaining connections in the masked NN averaged across layers, where 1.0 is equivalent to a fully connected network, and the test NLL when the flow is trained using N samples.


    | Factorization Method | Fraction remaining connections | Test NLL (N=100) | Test NLL (N=800) |
    | --- | --- | --- | --- |
    | Greedy | 0.255 | **15.765 ± 0.066** | **14.960 ± 0.115** |
    | Random [3] | 0.1245 | 16.733 ± 0.291 | 15.376 ± 0.183 |
    | Mincon | 0.085 | 17.607 ± 0.222 | 17.094 ± 0.078 |

    We also visualize the trend of these results as a function of dataset size (Figure 1 in the attached PDF). **We see clear improvement using the greedy approach, reiterating the importance of factorization with respect to an objective.**


Furthermore, we clarify **common concerns** the reviewers have raised:

1. **Comparisons to MADE**: Reviewers noted that we drew inspiration from MADE to mask weight matrices. We believe **our extension of this idea to enforce a specific, sparse adjacency structure is novel**. Further, a significant contribution is our observation that the **objective of the mask factorization is directly linked to NN architecture, which can affect performance**, as demonstrated in our experiments above.
2. **Alternative methods for enforcing structure (such as GNF):** Reviewers noted that other methods of enforcing a prescribed adjacency structure exist. Given D-dimensional input, one could use GNF, or initialize D independent neural networks, each accepting the dependent features for a specific variable. However, both these approaches require D forward passes to compute the output of a single input vector, whereas we only require one. **The lower asymptotic complexity of evaluating our model enables higher training efficiency** and is important when incorporating StrNN into density estimators such as CNFs (see above), which require many forward evaluations to compute a single ODE solution.
3. Several reviewers correctly noted that our current work on StrNN does not have the capability to **learn Bayesian networks from data**. **This was by design given the main assumption in our work was that we would have access to the ground truth DAG, whether from domain knowledge or prior experiments with structure discovery algorithms.** To direct readers to these resources, we will add citation to the **causal-learn package** in our final revision. We have further considered different ways we could utilize StrNN itself to learn structure from data, as discussed in our response to Reviewer Tj4q.

Finally, we would like to thank all reviewers once again for your insightful responses, which have helped us refine our work and identify future directions. We are confident our updated manuscript will have addressed your concerns and suggestions, and we kindly invite you to reconsider our paper if you are satisfied with our responses.

[1] Grathwohl, Will, et al. Ffjord: Free-form continuous dynamics for scalable reversible generative models. *International Conference on Learning Representations*, 2019.

[2] Weilbach, Christian, et al. "Structured conditional continuous normalizing flows for efficient amortized inference in graphical models." *International Conference on Artificial Intelligence and Statistics*. PMLR, 2020.

[3] Mouton, Jacobie, and Steve Kroon. "Graphical residual flows." *arXiv preprint arXiv:2204.11846* (2022).

---

### Decision · Program_Chairs · 2023-09-21

**Decision:**

Accept (poster)

**Comment:**

A good contribution to combine neural networks and graphical models such as Bayesian networks and causal models. The reviewers generally favored acceptance. Some concerns by the reviewers remain, which should be addressed by the authors.

The AC has some additional comments:

Since binary matrix factorization is NP-hard, this means that Algorithm 1 does not necessarily find a solution with the exact sparseness patterns prescribed by $A$, correct? As far as I get it, the Algorithm might introduce further zeros in $A'$, is this correct? The authors should please clarify this in the paper.

Please double check and enhance the description of Algorithm 1.

- First, it was not clear what line "Fill $M^W$ with $nz$; repeat until full" means. I only got it from Figure 7a in the appendix.
- In the visual example, perhaps start with $M^V$ indeed filled with ones, to see the effect of the algorithm.
- In line 4, we iterate $i$ over the rows of $M^V$. Then we look for the indices in the $i$-th row in $A$. However, A has d1 many rows and $M^V$ has d2 many rows, thus we potentially address into non-existing rows of $A$, if $d1 < d2$. (I suppose the algorithm should also work for $d1 \not= d2$?)